# Modular UBE2H-CTLH E2-E3 complexes regulate erythroid maturation

**Dawafuti Sherpa[1†], Judith Mueller[1†], Özge Karayel[2†], Peng Xu[3,4†], Yu Yao[4], Jakub Chrustowicz[1], Karthik V Gottemukkala[1], Christine Baumann[1], Annette Gross[1,5], Oliver Czarnecki[1‡], Wei Zhang[6§], Jun Gu[6,7], Johan Nilvebrant[6,7#], Sachdev S Sidhu[6,7], Peter J Murray[5], Matthias Mann[2], Mitchell J Weiss[4], Brenda A Schulman[1], Arno F Alpi[1\*]**

[1]Department of Molecular Machines and Signaling, Max Planck Institute of Biochemistry, Martinsried, Germany; [2]Department of Proteomics and Signal Transduction, Max Planck Institute of Biochemistry, Martinsried, Germany; [3]Cyrus Tang Medical Institute, National Clinical Research Centre for Hematologic Diseases, Collaborative Innovation Centre of Hematology, State Key Laboratory of Radiation Medicine and Protection, Soochow University, Suzhou, China; [4]Department of Hematology, St. Jude Children's Research Hospital, Memphis, United States; [5]Department of Immunoregulation, Max Planck Institute of Biochemistry, Martinsried, Germany; [6]Donnelly Centre for Cellular and Biomolecular Research, University of Toronto, Toronto, Canada; [7]Department of Molecular Genetics, University of Toronto, Toronto, Canada

**\*For correspondence:**
aalpi@biochem.mpg.de

[†]These authors contributed equally to this work

**Present address:** [‡]Institute of Diabetes and Regeneration Research, Helmholtz Centre Munich, Neuherberg, Germany; [§]Department of Molecular and Cellular Biology, College of Biological Science, University of Guelph, Guelph, Canada; [#]Division of Protein Engineering, School of Chemistry, Biotechnology and Health, Royal Institute of Technology, Stockholm, Sweden

**Abstract** The development of haematopoietic stem cells into mature erythrocytes – erythropoiesis – is a controlled process characterized by cellular reorganization and drastic reshaping of the proteome landscape. Failure of ordered erythropoiesis is associated with anaemias and haematological malignancies. Although the ubiquitin system is a known crucial post-translational regulator in erythropoiesis, how the erythrocyte is reshaped by the ubiquitin system is poorly understood. By measuring the proteomic landscape of in vitro human erythropoiesis models, we found dynamic differential expression of subunits of the CTLH E3 ubiquitin ligase complex that formed maturation stage-dependent assemblies of topologically homologous RANBP9- and RANBP10-CTLH complexes. Moreover, protein abundance of CTLH's cognate E2 ubiquitin conjugating enzyme UBE2H increased during terminal differentiation, and UBE2H expression depended on catalytically active CTLH E3 complexes. CRISPR-Cas9-mediated inactivation of CTLH E3 assemblies or UBE2H in erythroid progenitors revealed defects, including spontaneous and accelerated erythroid maturation as well as inefficient enucleation. Thus, we propose that dynamic maturation stage-specific changes of UBE2H-CTLH E2-E3 modules control the orderly progression of human erythropoiesis.

## Editor's evaluation

This paper will be of interest to scientists in the field of hematology and ubiquitin biology. The work identifies previously unrecognized functions of and regulatory mechanisms impinging on CTLH E3 ubiquitin ligases during erythrocyte progenitor maintenance and differentiation. It provides new insights into the dynamic formation of E3 ubiquitin ligases during development, suggesting that rather than simply exchanging substrate adaptors, scaffolding proteins and collaborating E2 enzymes are also tightly regulated. The experiments are of high quality and a wealth of data supports the conclusions.

## Introduction

Cellular differentiation in multicellular organisms is often accompanied by programmed proteome reshaping and cellular reorganization to accomplish cell-type-specific functions. For instance, during myogenesis proliferative myoblasts undergo a differentiation programme with induction of specialized cytoskeletal proteins to form myofibrils in terminally differentiated myofibers (*Chal and Pourquie, 2017*; *Le Bihan et al., 2015*), whereas adipose stem cells induce differentiation cues controlling expression of proteins involved in lipid storage and lipid synthesis (*Tsuji et al., 2014*). Recently, global temporal proteomic analysis during neurogenesis of human embryonic stem cells revealed large-scale proteome and organelle remodelling via selective autophagy (*Ordureau et al., 2021*). A striking example of proteome remodelling is mammalian erythropoiesis, which is required for the generation of disc-shaped enucleated erythrocytes, whose unique topology dictates function of efficient red blood cell movement through the vasculature (*Figure 1A*). After several specialized cell divisions, erythroid progenitors progress through morphologically distinct differentiation stages known as pro-erythroblasts (ProE), early and late basophilic erythroblasts (EBaso and LBaso, respectively), polychromatic erythroblasts (Poly), and orthochromatic erythroblasts (Ortho), a process associated with erythroid-specific gene expression (*Cantor and Orkin, 2002*; *Cross and Enver, 1997*; *Perkins et al., 1995*; *Pevny et al., 1991*; *Shivdasani et al., 1995*), reduction of cell volume (*Dolznig et al., 1995*), chromatin condensation (*Zhao et al., 2016*), and haemoglobinization. Ejection of the nucleus at the reticulocyte stage (*Keerthivasan et al., 2011*) is followed by the elimination of all remaining organelles such as Golgi, mitochondria, endoplasmic reticulum, peroxisomes, and ribosomes (*Moras et al., 2017*; *Nguyen et al., 2017*). The progression of erythroid maturation must be tightly controlled, although the molecular regulation of this process is not fully understood.

Our current knowledge of protein dynamics during erythropoiesis has been deduced largely from epigenetic and transcriptomic studies (reviewed in *An et al., 2015*), which used in vitro differentiation systems where erythroid progenitors, such as primary multipotent CD34[+] haematopoietic stem and progenitor cells (HSPC) or immortalized CD34[+]-derived lines (known as HUDEP2 and BEL-A) (*Kurita et al., 2013*; *Trakarnsanga et al., 2017*), possess an autonomous differentiation programme with a capacity to complete terminal differentiation when cultured with cytokines and other factors (*Seo et al., 2019*). However, the dynamics of mRNA expression during erythropoiesis does not accurately predict protein expression (*Gautier et al., 2016*). The erythroid proteome landscape of defined precursors generated by in vitro differentiation of normal donor CD34[+] cells was recently mapped (*Karayel et al., 2020*; *Peng et al., 2022*; *Gautier et al., 2016*), and the findings provided insight into cellular remodelling at protein resolution and indicated high level of post-transcriptional regulation.

Ubiquitin (UB)-mediated protein degradation pathways are likely to play prominent roles in post-transcriptional regulation of erythropoiesis. Ubiquitin conjugating enzymes (E2), components of the E3 ubiquitin ligase machinery, and deubiquitylases have been implicated in regulating protein stability and turnover in erythroid cell proliferation and maturation (*Feng et al., 2022*; *Liang et al., 2019*; *Maetens et al., 2007*; *Mancias et al., 2015*; *Minella et al., 2008*; *Nguyen et al., 2017*; *Randle et al., 2015*; *Thom et al., 2014*; *Xu et al., 2020*). The E2 enzyme UBE2O is greatly upregulated in reticulocytes and required for clearing ribosomes (*Nguyen et al., 2017*). Recently, a functional role of the multiprotein *C*-terminal to *LisH* (CTLH) E3 ubiquitin ligase was implicated in mammalian erythropoiesis. The CTLH subunits MAEA and WDR26 are expressed in a differentiation stage-dependent manner and implicated in maintaining erythroblastic islands in the bone marrow and regulating nuclear condensation in developing erythroblasts, respectively (*Wei et al., 2019*; *Zhen et al., 2020*). The tight correlation between protein abundance and functionality in differentiation suggested that large-scale proteome profiling is a potential way to identify proteins that are important for the functional specialization of erythroid cells. Here, we profiled protein abundance of E2-E3 modules in erythroid differentiation uncovering a dynamic regulation of CTLH E3 ligase subunits and its cognate E2 conjugating enzyme UBE2H. Interestingly, UBE2H amounts are dependent on active CTLH E3, suggesting a coupled E2/E3 regulation. We further show that CTLH complex composition is remodelled and complex assemblies are formed in a maturation stage-dependent manner. Our study indicates that unique UBE2H-CTLH assemblies are organized and co-regulated in functional E2-E3 modules and are required for the orderly progression of terminal erythroid maturation.

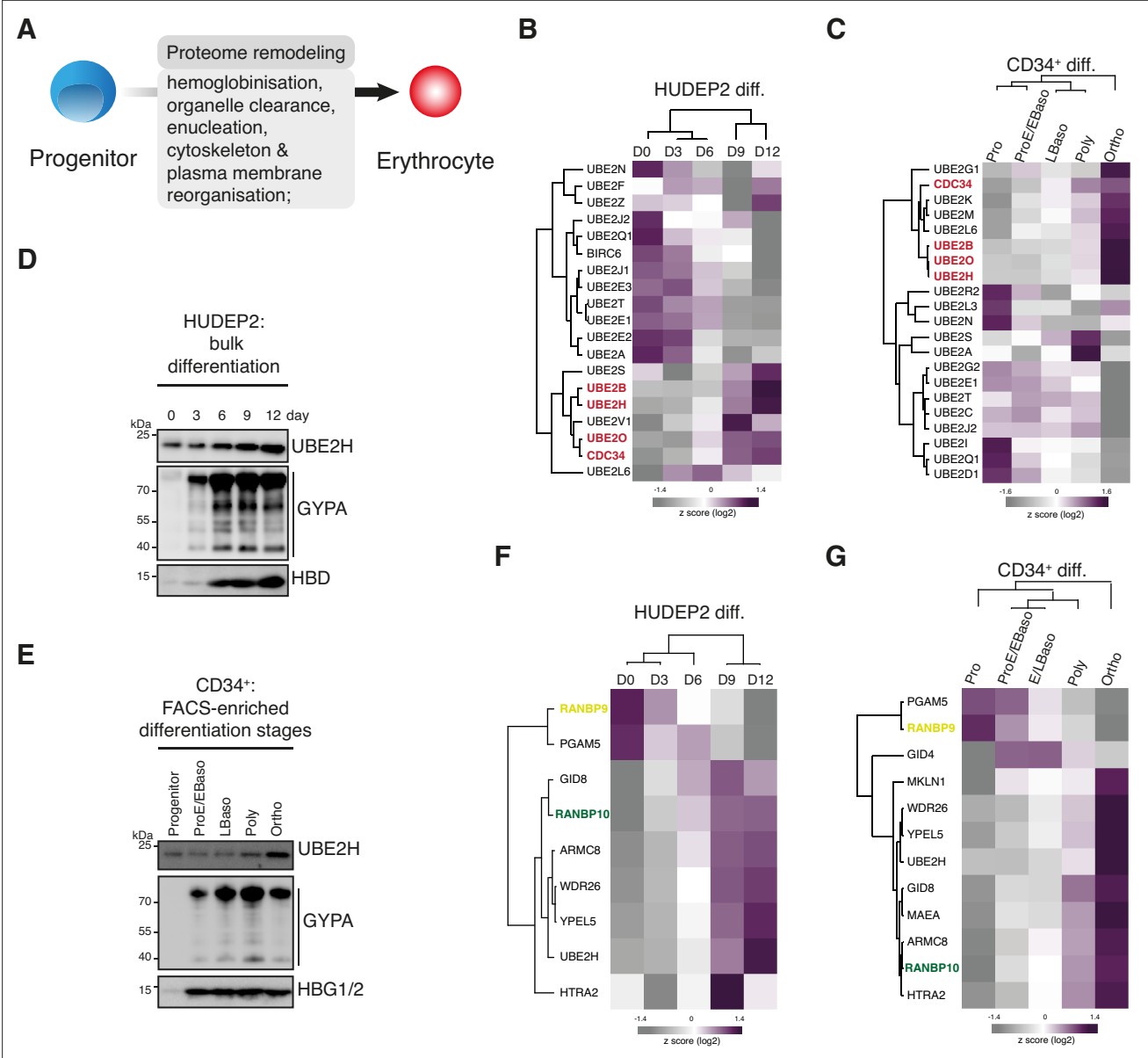

**Figure 1.** Stage-dependent expression of UBE2H and CTLH complex subunits during erythropoiesis. (**A**) Cartoon indicating key features of mammalian erythropoiesis. (**B**) Heat map of z-scored protein abundance (log$_2$ data-independent acquisition (DIA) intensity) of differentially expressed E2 enzymes in differentiated HUDEP2 cells. (**C**) Heat map of z-scored protein abundance (log$_2$ DIA intensity) of differentially expressed E2 enzymes in differentiated CD34$^+$ cells. (**D**) HUDEP2 cells were differentiated in vitro and analysed by immunoblotting with indicated antibodies. (**E**) CD34$^+$ cells were differentiated in vitro, cell populations enriched by FACS, and analysed by immunoblotting with indicated antibodies. (**F**) Heat map of z-scored protein abundance (log$_2$ DIA intensity) of differentially expressed CTLH complex subunits in differentiated HUDEP2 cells. (**G**) Heat map of z-scored protein abundance (log$_2$ DIA intensity) of differentially expressed CTLH complex subunits in differentiated CD34$^+$ cells.

The online version of this article includes the following source data and figure supplement(s) for figure 1:

**Source data 1.** Original, uncropped scans of immunoblots.

**Figure supplement 1.** Determination of differentiation stage-dependent proteomes from in vitro differentiated HUDEP2 cells.

## Results

### Stage-dependent expression of UBE2H and CTLH complex subunits during erythropoiesis

Reshaping of the erythropoietic proteome is thought to be regulated in part by the transient presence of stage-specific E2-E3 ubiquitin targeting machineries. To identify potential E2-E3

components, we applied a system-wide approach and established differentiation stage-specific proteomes of human erythropoiesis from two in vitro erythropoiesis cell model systems: CD34+ and HUDEP2 cells (*Figure 1—figure supplement 1A and B*). We recently described the stage-specific proteomes from in vitro differentiated CD34+ cells (*Karayel et al., 2020*). HUDEP2 cells proliferate in immature progenitor state and can be induced to undergo terminal erythroid differentiation by modulating cell culture conditions (*Figure 1—figure supplement 1B*; *Kurita et al., 2013*). In this study, HUDEP2 cells were shifted to differentiation conditions and semi-synchronous bulk cell populations were obtained at different time points (days 0, 3, 6, 9, and 12) corresponding to maturation stages spanning from proerythroblast to orthochromatic stages. Each population was processed in three biological replicates, and their proteomes were acquired by measuring single 100 min gradient runs for each sample/replicate in data-independent acquisition (DIA) mode (*Aebersold and Mann, 2016*; *Gillet et al., 2012*; *Karayel et al., 2020*; *Ludwig et al., 2018*). DIA raw files were searched with direct DIA (dDIA), yielding 6727 unique proteins and quantitative reproducibility with Pearson correlation coefficients greater than 0.9 between the biological replicates of all populations (*Figure 1—figure supplement 1C and D*). When we clustered the 2771 differentially expressed proteins (ANOVA, false discovery rate [FDR] < 0.01 and S0 = 0.1), we observed dynamic changes of the proteome between early (day 0) and late (day 12) time points across erythroid differentiation (*Figure 1—figure supplement 1E*). The majority of proteins cluster into two co-expression profiles: continuous decrease or increase of protein levels that ultimately resulted in a reshaped erythrocyte-specific proteome.

We next examined our proteome data for all (~40) annotated human E2 ubiquitin conjugating enzymes. The levels of most E2s detected in HUDEP2 cells varied across maturation, with a cluster of six enzymes progressively accumulating until day 12 (*Figure 1B*). Included among them was UBE2O, which mediates ribosomal clearance in reticulocytes (*Nguyen et al., 2017*). We expanded the analysis to stage-specific proteomes from in vitro differentiated CD34+ cells (*Figure 1C*; *Karayel et al., 2020*), which revealed a similar cluster of E2s upregulated at poly- and orthochromatic stages. Notably, UBE2B, UBE2O, CDC34 (aka UBE2R1), and UBE2H enzymes exhibited similar protein abundance profiles during HUDEP2 and CD34+ maturation, suggesting roles for these E2s during terminal erythropoiesis. CDC34, the cognate E2 for cullin-1 RING ligase (CRL1) complexes, is essential for cell cycle regulation (*Kleiger et al., 2009*; *Skaar and Pagano, 2009*). UBE2B (aka RAD6B) regulates DNA repair pathways, histone modifications, and proteasomal degradation (*Kim et al., 2009*; *Varshavsky, 1996*; *Watanabe et al., 2004*). We focused on UBE2H because it is transcriptionally regulated by the essential erythroid nuclear protein TAL1 and accumulates to high levels during terminal maturation (*Lausen et al., 2010*; *Nguyen et al., 2017*; *Wefes et al., 1995*). Immunoblotting confirmed UBE2H protein upregulation during maturation of HUDEP2 and CD34+ cells, which was paralleling induction of the erythroid membrane protein CD235a (glycophorin A [GYPA]) and haemoglobin expression (*Figure 1D and E*, *Figure 1—source data 1*).

The stage-dependent regulation of UBE2H suggested that a cognate E3 partnering with UBE2H would have a similar expression profile during erythropoiesis. In vitro ubiquitylation reactions indicate that UBE2H is the preferred E2 of the CTLH E3 ubiquitin ligase (*Lampert et al., 2018*; *Sherpa et al., 2021*). The multiprotein CTLH complex, orthologue of the yeast GID complex, consists of at least RANBP9, and/or RANBP10 (yeast Gid1), TWA1 (yeast Gid8), ARMC8 (yeast Gid5), WDR26, and/or MKLN1 (yeast Gid7), the catalytic module – MAEA and RMND5A (yeast Gid9 and Gid2) that mediate ubiquitin transfer, and the substrate receptor GID4 (yeast Gid4) (*Kobayashi et al., 2007*; *Lampert et al., 2018*; *Liu and Pfirrmann, 2019*; *Maitland et al., 2022*; *Mohamed et al., 2021*; *Salemi et al., 2017*; *Sherpa et al., 2021*; *Umeda et al., 2003*). Our analyses of the stage-dependent proteomes of differentiated HUDEP2 (*Figure 1F*) and CD34+ cells (*Figure 1G*) revealed that protein levels of most annotated CTLH subunits increased during erythroid maturation, in parallel with UBE2H. Interestingly, the homologues RANBP9 and RANBP10 showed an inverse expression pattern: RANBP9 levels were high at progenitor stages and dropped at later stages, whereas RANBP10 levels exhibited the opposite pattern. Taken together, analyses of differentiation-resolved proteomes revealed stage-dependent expression of CTLH subunits and UBE2H suggesting a dynamic assembly of distinct CTLH complexes linked to erythrocyte development.

## Erythroid maturation stage-dependent modulation of RANBP9- and RANBP10-assembled CTLH complexes

Recent cryo-EM maps of human CTLH sub- and supramolecular complexes revealed that RANBP9 is part of a core scaffold module of the CTLH complex (*Figure 2A*; *Sherpa et al., 2021*). Beyond an N-terminal extension unique to RANBP9, both homologues RANBP9 and RANBP10 have a common domain architecture (*Figure 2—figure supplement 1*). Hence, we reasoned that RANBP10 may replace RANBP9 and also form topologically similar complexes and, depending on abundance and availability of RANBP9 and RANBP10, distinct RANBP9-, 'mixed' RANBP9/RANBP10-, and/or RANBP10-CTLH complexes may assemble (*Figure 2B*). To test this hypothesis, we monitored CTLH complexes by fractionating whole-cell lysates from non-differentiated (day 0) or differentiated (day 6) HUDEP2 cells on 5–40% sucrose density gradients and detecting CTLH subunits by immunoblot analysis. All CTLH subunits sedimented at ≥670 kDa, corresponding to the shift observed for the supramolecular CTLH assemblies we previously described (*Figure 2C*, *Figure 2—source data 1*; *Sherpa et al., 2021*). However, RANBP9 amounts in the CTLH fraction were higher at day 0 compared to day 6, while RANBP10 amounts had the opposite pattern, suggesting stage-specific modulation of CTLH complex composition and/or stoichiometry during differentiation. To further test stage-specific modulation of CTLH, we established a sequential immunoprecipitation (IP) protocol allowing us to determine relative proportions of RANBP9-, RANBP9/RANBP10-, and RANBP10-CTLH complexes in cell lysates (*Figure 2D*). In step 1, RANBP9-assembled complexes were immunoprecipitated from lysates of differentiation days 0, 4, and 8 using RANBP9-specific antibody. Subsequently, RANBP9-depleted supernatants were subjected to step 2 for immunoprecipitation with an ARMC8-specific nanobody (*Figure 2—figure supplement 2*) to precipitate remaining RANBP10-CTLH complexes. Immunoblot analysis of pellet 1 samples revealed a progressive decrease of precipitated RANBP9-CTLH in differentiating HUDEP2 cells (*Figure 2E*, *Figure 2—source data 2*). Notably, RANBP10 was co-precipitated and relative RANBP10 amounts increased towards differentiation day 8, indicating a shift from RANBP9- to RANBP9/RANBP10-assembled CTLH complexes. Moreover, the amount of precipitated RANBP10-CTLH complexes in pellet 2 samples increased, indicating the predominant assembly of RANBP10-CTLH complexes at late stages of differentiation (*Figure 2B and E*). To further test whether RANBP9 and RANBP10 can independently form CTLH complexes, we deleted either *RANBP9* or *RANBP10* in HUDEP2 cells using CRISPR-Cas9 editing (*Figure 2F*, *Figure 2—source data 2*, *Figure 2—figure supplement 3*). Whole-cell lysates of parental and KO lines were analysed using sucrose density gradients and immunoblot analysis to assess the sedimentation of RANBP9 and RANBP10 along with other CTLH subunits WDR26 and MAEA (*Figure 2G*, *Figure 2—source data 3*). The sedimentation of the supramolecular CTLH complex containing RANBP9 was similar in parental and *RANBP10*[-/-] cells. Likewise, RANBP10-CTLH assemblies sedimented similarly in parental and *RANBP9*[-/-] cells. These data indicate a dynamic modulation of RANBP9- to RANBP10-assembled supramolecular CTLH complexes during the process of erythroid maturation.

## RANBP9 and RANBP10 form similar CTLH complex structures that cooperate with UBE2H to promote ubiquitin transfer

As RANBP9 and RANBP10 can independently assemble in supramolecular CTLH complexes in cells, we next asked whether RANBP10 forms a similar molecular CTLH structure as described previously for RANBP9 (*Sherpa et al., 2021*). To test this, we expressed and purified a recombinant version of the core CTLH subcomplex, containing a scaffold module (RANBP10, TWA1, α-ARMC8), the catalytic module (MAEA and RMND5A), and the substrate receptor GID4 (named hereafter RANBP10-CTLH[SR4]). In parallel, we generated the previously described homologous complex where RANBP9 replaced RANBP10 (RANBP9-CTLH[SR4]) (*Mohamed et al., 2021*; *Sherpa et al., 2021*). The two complexes eluted at similar range in size-exclusion chromatography (SEC), indicating they had comparable subunit stoichiometry (*Figure 3A*, *Figure 3—source data 1*). Cryo-EM analysis of the RANBP10-CTLH[SR4] peak fraction yielded a reconstitution at ~12 Å resolution (EMDB: EMD-16242) (*Figure 3B*). Comparison to the previously determined RANBP9-CTLH[SR4] map (EMDB: EMD-12537) revealed overall structural similarity, including the clamp-like assembly of substrate receptor scaffolding (SRS) and catalytic (Cat) modules conserved in related yeast GID complexes, albeit with differences in the extent of the catalytic modules visible in the maps (*Qiao et al., 2020*; *Sherpa et al., 2021*; *Figure 3B*, *Figure 3—figure supplement 1*). Importantly, atomic coordinates of α-ARMC8, GID4, and TWA1 derived from the

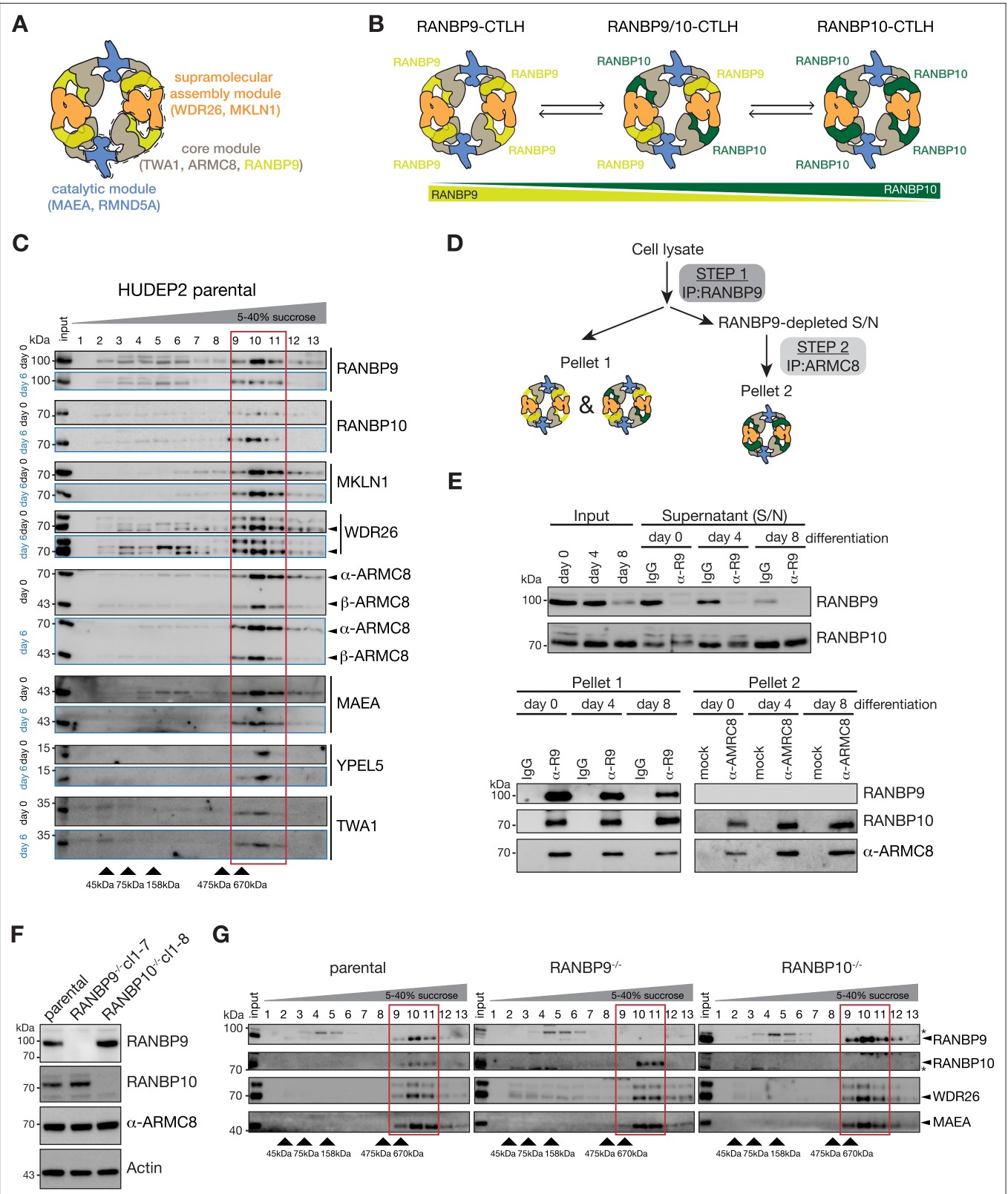

**Figure 2.** Erythroid maturation stage-dependent modulation of RANBP9- and RANBP10-assembled CTLH complexes. (**A**) Cartoon of the supramolecular RANBP9-CTLH assembly indicating the catalytic (blue), core (grey/yellow), and supramolecular assembly (orange) modules. (**B**) Model of remodelling RANBP9- and RANBP10-CTLH complexes. (**C**) HUDEP2 cell lysates from differentiation days 0 and 6 were separated on sucrose gradients, and fractions analysed by immunoblotting with indicated antibodies. Fractions containing supramolecular CTLH assemblies are boxed in

*Figure 2 continued on next page*

*Figure 2 continued*

red. (**D**) Workflow of the sequential immunoprecipitation (IP) to determine RANBP9- and RANBP10-CTLH complexes. (**E**) HUDEP2 cell lysates from differentiation days 0, 4, and 8 were subjected to sequential IPs with RANBP9 antibody and ARMC8-specific nanobody as described in (**D**) followed by immunoblot analysis with indicated antibodies. IgG: unspecific antibody; mock: absence of nanobody. (**F**) Immunoblots of lysates of HUDEP2 parental, RANBP9$^{-/-}$, and RANBP10$^{-/-}$ cells probing for RANBP9 and RANBP10. Actin serves as loading control. (**G**) Sucrose gradient fractionation of HUDEP2 cell lysates from RANBP9$^{-/-}$ or RANBP10$^{-/-}$ knock out lines, fractions were analysed by immunoblotting with indicated antibodies. Fractions containing supramolecular CTLH assemblies are boxed in red.

The online version of this article includes the following source data and figure supplement(s) for figure 2:

**Source data 1.** Original, uncropped scans of immunoblots.

**Source data 2.** Original, uncropped scans of immunoblots.

**Source data 3.** Original, uncropped scans of immunoblots.

**Figure supplement 1.** Sequence and structural alignments of human RANBP9 and RANBP10.

**Figure supplement 2.** Cryo-EM structure of the ARMC8-specific nanobody bound to CTLH core (EMDB: EMD-16243).

**Figure supplement 3.** Schematics of CRISPR-Cas9 edited knockouts of RANBP9 and RANBP10.

RANBP9-CTLH$^{SR4}$ structure (PDB: 7NSC), along with crystal structure of RANBP10-SPRY domain (PDB: 5JIA), fit into the 7.6 Å resolution-focused refined map of RANBP10-CTLH$^{SR4}$ (*Figure 3C*). To position RANBP10, the crystal structure of the RANBP10's SPRY domain (PDB: 5JIA; *Hong et al., 2016*) was superimposed to the structure of the RANBP9's SPRY domain from recently published RANBP9-CTLH$^{SR4}$ (PDB: 7NSC; *Sherpa et al., 2021*; *Figure 2—figure supplement 1B*). Thus, at an overall level, the core RANBP10-CTLH$^{SR4}$ and RANBP9-CTLH$^{SR4}$ complexes are structurally homologous.

We next asked whether the structural similarity of the core RANBP10-CTLH$^{SR4}$ and RANBP9-CTLH$^{SR4}$ complexes extended to the mechanism of ubiquitin transfer activity. First, we assessed the physical association between UBE2H and CTLH complex subunits in HUDEP2 or erythroleukemia K562 cells using anti-UBE2H IPs (*Figure 3D and E*, *Figure 3—source data 1*; *Andersson et al., 1979*). Endogenous UBE2H specifically co-precipitated the Cat-module subunit MAEA in whole-cell lysates from both cell lines, indicating that UBE2H can form a reasonably stable E2-E3 enzyme module. Next, we tested ubiquitin transfer activity by in vitro ubiquitylation assays with a fluorescently labelled model peptide substrate (*Sherpa et al., 2021*). This model substrate consisted of an N-terminal PGLW sequence that binds human GID4 (*Dong et al., 2020*; *Dong et al., 2018*) and a 30 residue linker sequence with the target lysine (K) towards the C-terminus at position 23 (PGLW[X]$_{30}$K23) (*Figure 3F*, *Figure 3—source data 1*). In a reaction with UBE2H, both RANBP9-CTLH and RANBP10-CTLH promoted polyubiquitylation of the model substrate peptide in a GID4-dependent manner, although RANBP10-CTLH was less active under these conditions than the homologous RANBP9-CTLH complex. Cumulatively, biochemical and structural data revealed that RANBP9 and RANBP10 can assemble in distinct homologous CTLH complexes capable of activating UBE2H-dependent ubiquitin transfer activity. More broadly, CTLH may represent a larger family of E3 ligase complexes generated by assembly of different variable members with invariable core scaffold subunits.

## Catalytically inactive CTLH E3 complexes and UBE2H deficiency cause aberrant erythroid maturation

To investigate potential functional roles of CTLH E3 and UBE2H in erythroid maturation, we first used K562 cells as a surrogate erythroid cell model that expresses erythroid markers, including CD235a/ GYPA and haemoglobin upon treatment with the pan histone deacetylase inhibitor Na-butyrate (NaB) (*Andersson et al., 1979*). We generated K562 *UBE2H*$^{-/-}$ and *MAEA*$^{-/-}$ cells by CRISPR-Cas9 editing (*Figure 4—figure supplement 1*; *Sherpa et al., 2021*). Presumably, deletion of *MAEA*, part of the catalytic RING subunit in all CTLH E3 assemblies, would result in a complete loss of all CTLH E3 ligase activities. Parental and *MAEA* or *UBE2H* knockout cell lines were either mock- or NaB-treated, and erythroid maturation was assessed by maturation marker CD235a/GYPA surface expression via flow cytometry. At baseline cell conditions (mock), *MAEA*$^{-/-}$ and *UBE2H*$^{-/-}$ lines showed increased CD235a/ GYPA-positive (CD235a$^{+}$) cells comparable to NaB-treated parental cells (*Figure 4—figure supplement 2A*). Furthermore, immunoblot analysis showed increased CD235a/GYPA expression in whole-cell lysates of *MAEA*$^{-/-}$ and *UBE2H*$^{-/-}$ lines treated at low dose of NaB, suggesting that *MAEA* and *UBE2H* deficiency might promote erythroid differentiation (*Figure 4—figure supplement 2B and C*,

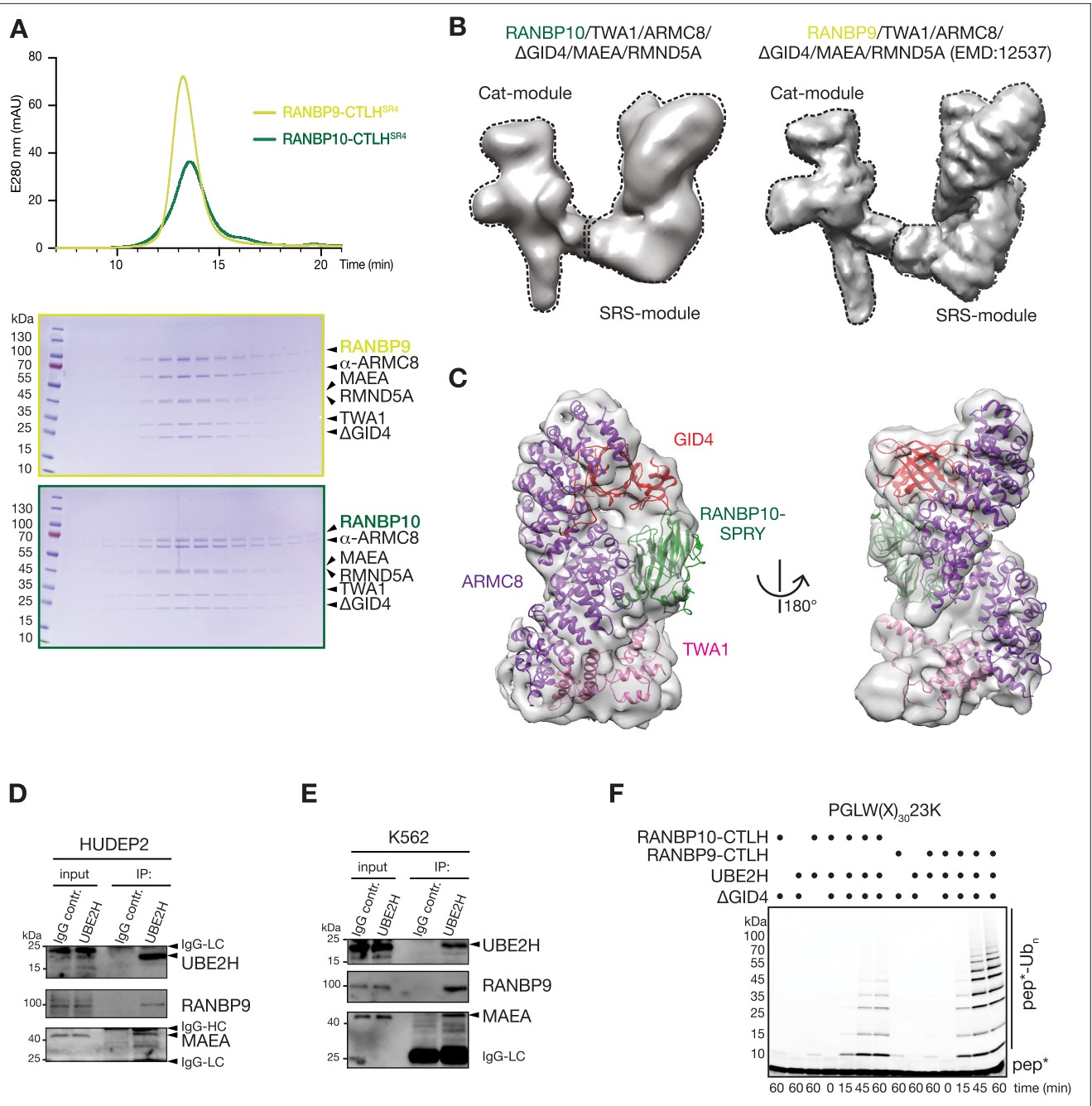

**Figure 3.** RANBP9 and RANBP10 form similar CTLH complex structures that cooperate with UBE2H to promote ubiquitin transfer. (**A**) Chromatograms (top) and Coomassie-stained SDS-PAGE gels (bottom) from size-exclusion chromatography of recombinant RANBP10-CTLH$^{SR4}$ and RANBP9-CTLH$^{SR4}$ complexes. (**B**) Cryo-EM map of RANBP10-CTLH$^{SR4}$ (EMDB: EMD-16242, left) and RANBP9-CTLH$^{SR4}$ (EMDB: EMD-12537) (right) with Cat-module and SRS-module indicated. (**C**) Focused refined map of the RANBP10-CTLH SRS-module with coloured subunits: ARMC8, purple; TWA1, salmon;GID4(Δ1–99), red; RANBP10 SPRY-domain, green. (**D**) Immunoprecipitation (IP) from HUDEP2 cell lysates with IgG control and UBE2H-specific antibody and immunoblot analysis. (**E**) IP from K562 cell lysates with IgG control and UBE2H-specific antibody and immunoblot analysis. IgG light chain (IgG-LC), IgG heavy chain (IgG-HC). (**F**) Fluorescence scan of SDS-PAGE gels presenting time course of in vitro ubiquitylation assay with fluorescently labelled model substrate peptide PGLW(X)$_n$-23K with lysine at position 23 (pep*) in the presence of UBE2H, RANBP10-CTLH or RANBP9-CTLH, and GID4.

The online version of this article includes the following source data and figure supplement(s) for figure 3:

**Source data 1.** Original, uncropped scans of Coomassie-stained SDS-PAGEs and immunoblots.

**Figure supplement 1.** Flowchart of cryo-EM processing for the RANBP10-CTLH complex dataset.

*Figure 4—figure supplement 2—source data 1*). To further validate the observation, we expanded the analysis to *MAEA*- and *UBE2H*-deficient HUDEP2 cell lines (*Figure 4A*, *Figure 4—figure supplement 1A and B*). Flow cytometry analysis revealed an elevated proportion of CD235$^+$ cells in clones lacking *MAEA* (cl3-1 and cl23) or *UBE2H* (cl13 and cl16), indicating spontaneous erythroid maturation in expansion medium (*Figure 4B*, *Figure 4—figure supplement 2D*). As terminal erythropoiesis is characterized by a stage-dependent proteome remodelling (*Gautier et al., 2016*; *Karayel et al., 2020*), we next asked whether spontaneous erythroid maturation of *UBE2H*$^{-/-}$ and *MAEA*$^{-/-}$ cells is in line with altered global proteomes. Global proteomic analysis of undifferentiated parental, *UBE2H*$^{-/-}$ (clone 13), and *MAEA*$^{-/-}$ (clone 3-1) HUDEP2 cells identified 6210 unique proteins in total (*Figure 1—figure supplement 1C and D*) (Student's *t*-test with FDR < 0.05 and S0 = 0.1). We found that 18% (1170) and 6.5% (404) of all proteins were significantly changed in *UBE2H*$^{-/-}$ vs. parental and *MAEA*$^{-/-}$ vs. parental comparisons, respectively (Student's *t*-test with FDR < 0.05 and S0 = 0.1) (*Figure 4—figure supplement 2E and F*). Notably, 271 of these proteins were differentially changed in both, a MAEA- and UBE2H-dependent manner. These included several erythroid-specific proteins including haemoglobin subunits (HBD, HBG2, and HBM) and erythroid maturation marker Band3 (SLC4A1) in both comparisons, indicating an erythroid-typical remodelled proteome (*Figure 4C*). In Gene Ontology (GO) enrichment analysis, proteins associated with annotations related to erythropoiesis such as 'haemoglobin complex' and 'oxygen binding' were significantly enriched in both *MAEA*$^{-/-}$ and *UBE2H*$^{-/-}$ cells compared to parental cells (*Figure 4D*).

The deregulated proteome landscapes of *UBE2H*$^{-/-}$ and *MAEA*$^{-/-}$ cells indicated a functional role of UBE2H-CTLH modules in the initiation of erythroid differentiation of progenitors and/or in the maintenance of the progenitor stage. Next, we induced erythroid maturation for 3 days and evaluated erythroid markers CD49d (integrin alpha 4) and Band3 (SLC4A1). Each *MAEA*$^{-/-}$ and *UBE2H*$^{-/-}$ clone showed higher CD49d$^+$/Band3$^+$ cell populations compared to parental HUDEP2, indicating either precocious or accelerated maturation (*Figure 4E*, *Figure 4—figure supplement 2H*). To determine whether *MAEA*$^{-/-}$ and *UBE2H*$^{-/-}$ cells mature faster compared to controls, we sorted Band3$^-$ cells to generate 'synchronous' non-differentiated populations prior to differentiation. CD49d$^+$/Band3$^+$ measurement after day 4 confirmed an accelerated maturation in the absence of UBE2H or MAEA (*Figure 4F and G*). We expanded the analysis to differentiation time-specific proteomes (days 0–12) of parental and *MAEA*$^{-/-}$ (clone 3-1) cells (*Figure 4—figure supplement 3A–C*). The five distinct temporal stages of erythroid differentiation clustered separately by principal component analysis (PCA) with high consistencies between the three biological replicates (*Figure 4—figure supplement 3D*). Remarkably, parental versus *MAEA*$^{-/-}$ clusters progressively diverged up to differentiation day 6 and remained separated to day 12, suggesting an MAEA-dependent proteome remodelling at early stages of differentiation. PCA based on 28 erythroid-specific marker proteins (*Figure 4—figure supplement 3C*) revealed that the separation of parental versus *MAEA*$^{-/-}$ cluster was pronounced even at early differentiation time points day 0 (*Figure 4H*). In fact, the *MAEA*$^{-/-}$ cluster at day 0 shows a closer correlation with the parental cluster at day 3 than day 0, indicating proteome-wide changes towards an erythroid-specific proteome signature.

The stage-wide altered global proteome of MAEA$^{-/-}$ cells indicated that other differentiation-associated processes might be affected. Given that *MAEA*$^{-/-}$ mice embryos are anaemic with nucleated erythrocytes in peripheral blood (*Soni et al., 2006*), we focused our investigations on enucleation. HUDEP2 cells show intrinsically weak enucleation efficiency (*Kurita et al., 2013*), hence we switched to the CD34$^+$ erythroid differentiation system. Human CD34$^+$ cells were targeted with either MAEA sgRNA- or UBE2H sgRNA-assembled ribonucleoproteins (RNPs) and subjected to erythroid differentiation (see 'Materials and methods' for details). Enucleation was assessed by determining the pool of Hoechst$^-$/CD235$^+$ cells at differentiation day 14. Depletion of MAEA (*Figure 4I*) or UBE2H (*Figure 4K*) caused a measurable reduction of enucleated erythroid cells (*Figure 4J and L*, *Figure 4—figure supplement 4*), indicating that both MAEA and UBE2H are required for efficient enucleation. Taken together, the data suggest that active UBE2H-CTLH modules are required for the timely and accurate progression of erythroid maturation.

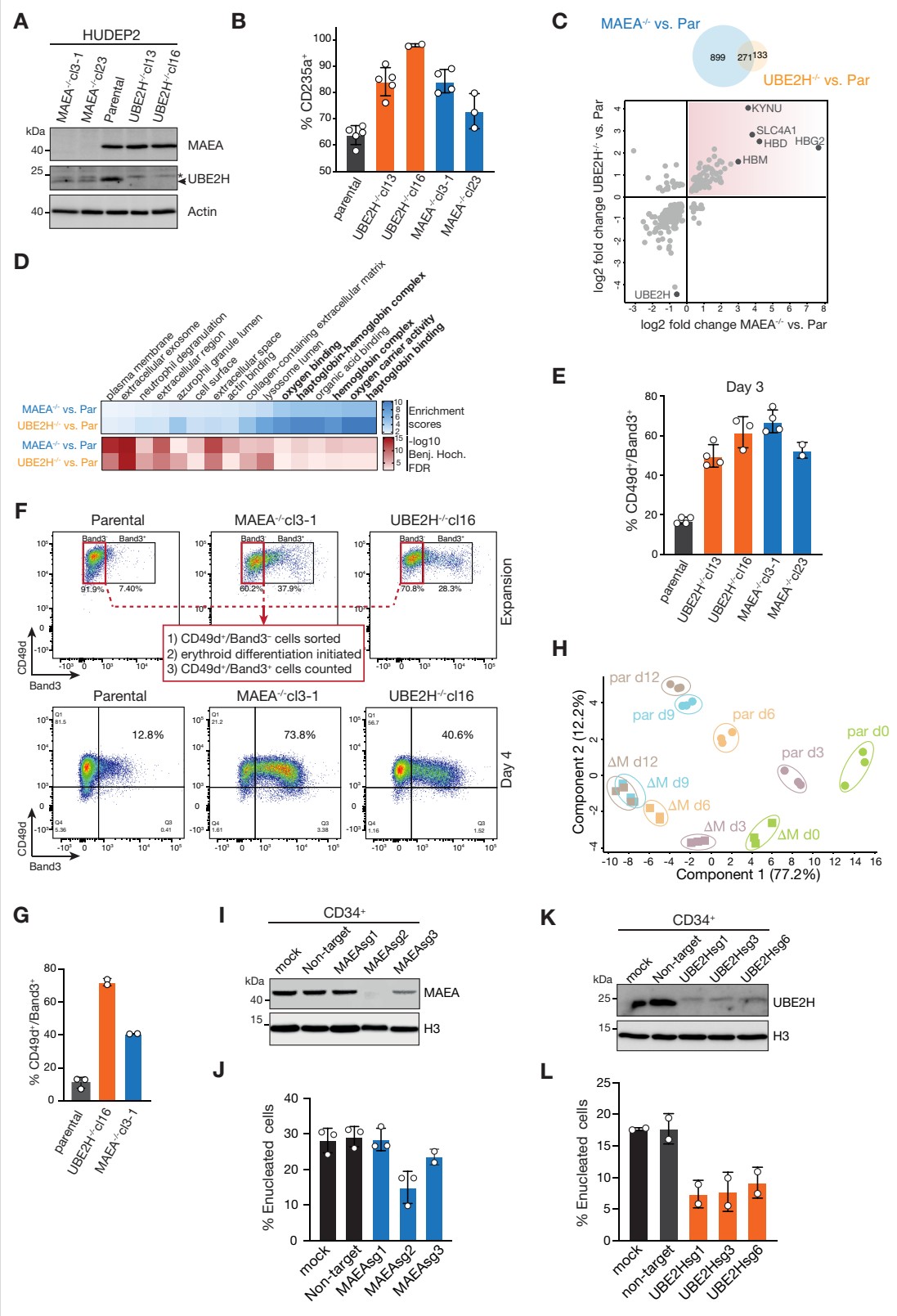

**Figure 4.** Catalytically inactive CTLH E3 complexes and UBE2H deficiency cause aberrant erythroid maturation. (**A**) Immunoblots of lysates of HUDEP2 parental, MAEA⁻/⁻, and UBE2H⁻/⁻ knockout clones probing for MAEA and UBE2H. Actin serves as loading control. (**B**) Quantitation of flow cytometry blots of indicated HUDEP2 cell lines assessing fraction of CD235a⁺ cells in expansion growing media condition. Error bars represent mean ± SD of n = 3–5 biological replicates. (**C**) Overlap between proteins with abundance differences of UBE2H⁻/⁻ versus parental and MAEA⁻/⁻ versus parental comparisons

*Figure 4 continued on next page*

*Figure 4 continued*

(top). Proteins with abundance differences (log$_2$) of UBE2H$^{-/-}$ versus parental blotted against proteins with abundance differences (log$_2$) MAEA$^{-/-}$ versus parental comparisons (bottom). Commonly enriched proteins are highlighted (top-right quadrant). (**D**) Gene Ontology (GO) enrichment analyses of upregulated protein in MAEA$^{-/-}$ versus parental and UBE2H$^{-/-}$ versus parental comparisons performed using Fisher's exact test (Benj. Hoch. FDR 5%). (**E**) Quantitation of flow cytometry blots of indicated HUDEP2 cell lines assessing fraction of CD49d$^+$/Band3$^+$ cells at differentiation day 3. Error bars represent mean ± SD of n = 2–4 biological replicates. (**F**) Indicated HUDEP2 cell lines, cultured in expansion media, were sorted for CD49d$^+$/Band3$^-$ (flow cytometry blots, top) followed by induction of erythroid maturation. Flow cytometry blots of indicated HUDEP2 cell lines showing CD49d/Band3 expression at day 4 after induced erythroid maturation. (**G**) Quantitation of (**H**) with graph showing fraction of CD49d$^+$/Band3$^+$ cells. Error bars represent mean ± SD of n = 2 biological replicates. (**H**) Principal component analysis (PCA) of erythroid differentiation stages (day 0, green; day 3, purple; day 6, orange; day 9, blue; day 12, brown) of HUDEP2 parental (par) and MAEA$^{-/-}$cl3-1 (ΔM) cell lines with their biological replicates based on expression profiles of selected erythroid marker proteins. (**I**) Immunoblot analysis of CRISPR-Cas9-mediated targeting of *MAEA* (MAEAsg1-3) in CD34$^+$ cells. Histone H3 serves as loading control. (**J**) Quantitation of flow cytometry blots of MAEAsg-targeted cell lines assessing fraction of enucleated cells. Error bars represent mean ± SD of n = 2 biological replicates. (**K**) Immunoblot analysis of CRISPR-Cas9-mediated targeting of *UBE2H* (UBE2Hsg1-3) in CD34$^+$ cells. Histone H3 serves as loading control. (**L**) Quantitation of flow cytometry blots of UBE2Hsg-targeted cell lines assessing fraction of enucleated cells. Error bars represent mean ± SD of n = 2 biological replicates.

The online version of this article includes the following source data and figure supplement(s) for figure 4:

**Source data 1.** Original, uncropped scans of immunoblots.

**Figure supplement 1.** Generation of CRISPR-Cas9 edited knockouts of UBE2H and MAEA.

**Figure supplement 1—source data 1.** Original, uncropped scans of immunoblots.

**Figure supplement 2.** Erythroid maturation analysis of MAEA- or UBE2H-deficient K562 and HUDEP2 cells.

**Figure supplement 2—source data 1.** Original, uncropped scans of immunoblots.

**Figure supplement 3.** Proteome landscape of in vitro differentiated parental and MAEA-deficient HUDEP2 cells.

**Figure supplement 4.** Enucleation analysis of MAEA- and UBE2H-deficient CD34+ cells during erythroid maturation.

## *RANBP9* but not *RANBP10* deficiency causes accelerated erythroid maturation

We next aimed to further dissect the function of distinct RANBP9- and RANBP10-CTLH complexes. Initially, we analysed pools of *RANBP9* and *RANBP10* CRISPR-Cas9-edited HUDEP2 cells for altered expression of CD235a/GYPA. Despite efficient depletion of RANBP9 or RANBP10 (***Figure 5A***, ***Figure 5—source data 1***), increased expression of the erythroid marker was only observed in RANBP9-depleted cells (***Figure 5B***, ***Figure 5—figure supplement 1A***). In agreement, isolated *RANBP9$^{-/-}$* clones (***Figure 5C***, ***Figure 5—source data 1***) showed increased CD235$^+$ populations, whereas *RANBP10$^{-/-}$* clonal lines (***Figure 5C***, ***Figure 5—source data 1***) express CD235a comparable to parental cells (***Figure 5D and E***, ***Figure 5—source data 1***, ***Figure 5—figure supplement 1B***). Next, erythroid maturation was induced in these deletion cell lines and progression evaluated by assessing CD49d$^+$/Band3$^+$ populations at day 3. Each *RANBP9$^{-/-}$* clone showed higher CD49d$^+$/Band3$^+$ cell populations compared to parental HUDEP2, indicating accelerated maturation as observed in *UBE2H*- and *MAEA*-deficient cells (***Figure 5F and G***). By contrast, *RANBP10$^{-/-}$* clones showed differentiation progression comparable to parental cells (***Figure 5H and I***). These data indicated that RANBP9- and RANBP10-CTLH complexes are required at different stages of differentiation and further suggest a role of RANBP9-CTLH in either maintaining HUDEP2 cells in a dormant/quiescent progenitor stage or controlling transition to erythroid differentiation.

## Cellular abundance of UBE2H is coupled to functional MAEA

Evidence for a regulatory relationship between UBE2H and MAEA during erythropoiesis was provided by an unexpected observation that UBE2H protein amounts are dependent on MAEA. Proteomics and immunoblot analyses showed consistently lower UBE2H in K562 and HUDEP2 *MAEA$^{-/-}$* cell lines (***Figure 4A and C***). Moreover, differentiating *MAEA$^{-/-}$* cells failed to express increased UBE2H protein levels at terminal maturation stages (days 9 and 12) (***Figure 6A***, ***Figure 6—source data 1***). The UBE2H mRNA, however, was not significantly different between parental and *MAEA$^{-/-}$* cells at day 9 of differentiation (***Figure 6B***), indicating that transcriptional regulation of UBE2H is not affected. We next expanded our analysis to K562 cells (***Figure 6C***). Parental and knockout K562 cell lines were either treated with NaB to induce erythroid-like or with 12-O-tetradodecanoyl-phorbol-13 acetate (TPA, chemical activator of PKC kinase) to induce megakaryocyte-like differentiation (***Tabilio et al., 1983***).

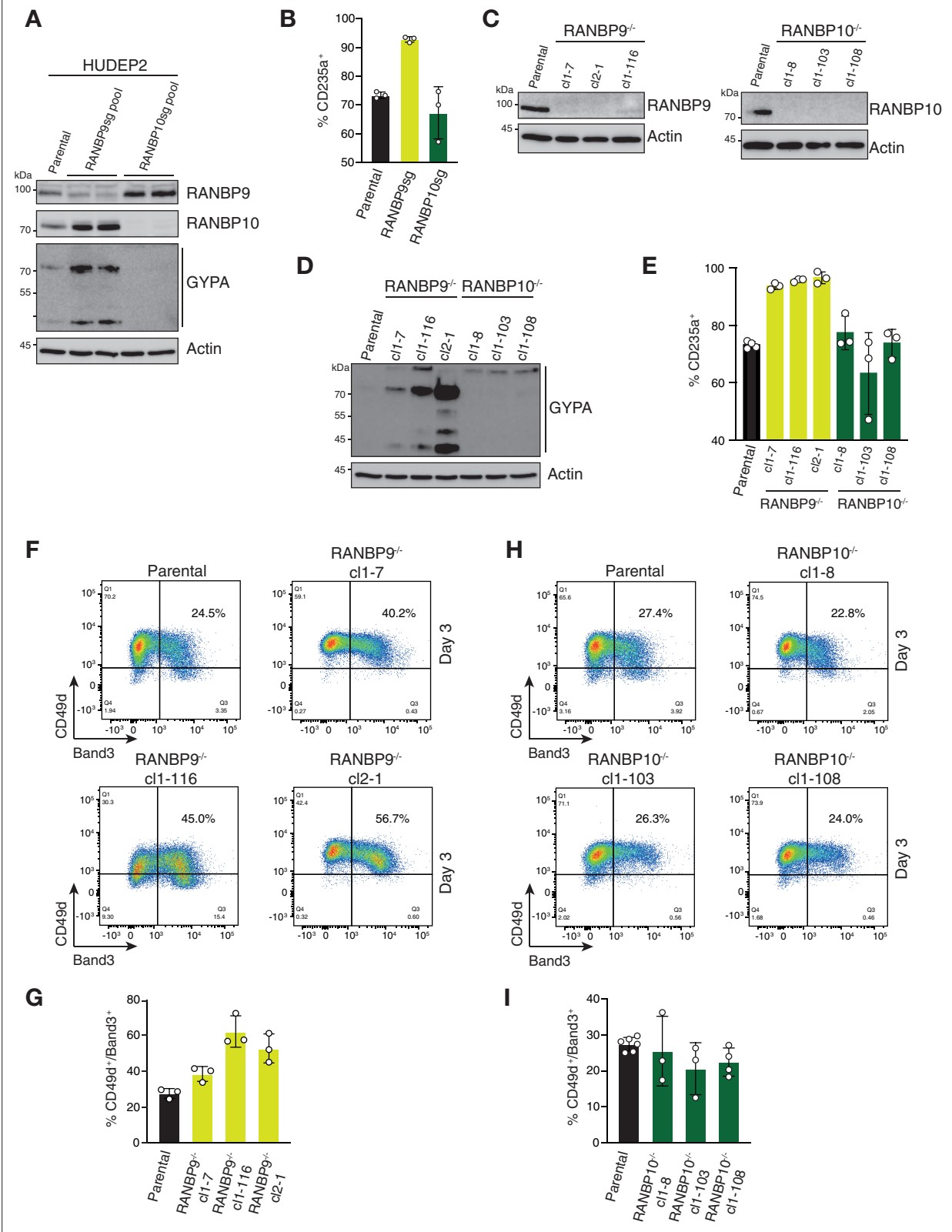

**Figure 5.** *RANBP9* but not *RANBP10* deficiency causes accelerated erythroid maturation. (**A**) Immunoblot analysis of CRISPR-Cas9-edited (RANBP9 or RANBP10) HUDEP2 cell pools using indicated antibodies. (**B**) Quantitation of flow cytometry blots of indicated CRISPR-Cas9 HUDEP2 cell pools assessing fraction of CD235a⁺ cells in expansion growing media condition. Error bars represent mean ± SD of n = 3 biological replicates. (**C, D**) Immunoblot analysis of HUDEP2 *RANBP9-/-* or *RANBP10⁻/⁻* cell lines using indicated antibodies. (**E**) Quantitation of flow cytometry blots of indicated

*Figure 5 continued on next page*

*Figure 5 continued*

CRISPR-Cas9 HUDEP2 cell lines assessing fraction of CD235a⁺ cells in expansion growing media condition. Error bars represent mean ± SD of n = 3 biological replicates. (**F, H**) Flow cytometry blots of indicated HUDEP2 cell lines showing CD49d/Band3 expression at day 3 after induced erythroid maturation. (**G, I**) Quantitation of (**F**) and (**H**) with graphs showing fraction of CD49d⁺/Band3⁺ cells. Error bars represent mean ± SD of n = 3 biological replicates.

The online version of this article includes the following source data and figure supplement(s) for figure 5:

**Source data 1.** Original, uncropped scans of immunoblots.

**Figure supplement 1.** CD235a expression of RANBP9- and RANBP10-deficient HUDEP2 cells.

Both NaB and TPA efficiently induced UBE2H protein levels in parental cells, suggesting that UBE2H regulation is not restricted to erythroid differentiation (*Figure 6D and E*, *Figure 6—source data 1*). By contrast, *MAEA⁻ᐟ⁻* cells had constitutively less and only marginally induced UBE2H protein amounts in response to NaB or TPA. Notably, cells either lacking CTLH's substrate receptor GID4 or subunits of the supramolecular module, WDR26 and MKLN1, showed UBE2H abundance and regulation similar to parental cells (*Figure 6D and E*, *Figure 6—figure supplement 1*, *Figure 6—figure supplement 1—source data 1*). The assessment of UBE2H mRNA levels revealed no significant difference of NaB-induced UBE2H transcription between parental and *MAEA⁻ᐟ⁻* cells (*Figure 6F*).

To further investigate the mechanism that underlies regulation of UBE2H and stability, we first asked whether UBE2H is targeted by proteasomal degradation. K562 parental and *MAEA⁻ᐟ⁻* cells were mock- or NaB-treated followed by a time-course treatment with proteasomal inhibitor (MG132). Whereas MG132 treatment for 2 hr had only a modest effect on UBE2H amounts, 6–24 hr exposure resulted in an increasing stabilization of UBE2H in *MAEA⁻ᐟ⁻* cells (lanes 6 and 8) matching parental cells (lanes 2 and 4) (*Figure 6G and H*, *Figure 6—source data 2*). Notably, NaB-induced UBE2H levels are comparable to 24 hr MG132-treated parental cells. Second, we asked whether MAEA activity is required for UBE2H abundance and stability. To this end, we used the MAEA Y394A mutation (MAEA-Y394A) that abolishes activity of the catalytic module in vitro (*Sherpa et al., 2021*), but maintains binding capacity to UBE2H in IP experiments (*Figure 6—figure supplement 1*, *Figure 6—figure supplement 1—source data 1*). K562 *MAEA⁻ᐟ⁻* cells stably expressing Flag-tagged MAEA-Y394A did hardly rescue UBE2H protein amounts, whereas wildtype MAEA expression resulted in a robust increase of UBE2H amounts (*Figure 6I and J*, *Figure 6—source data 2*). MAEA and RMND5A are structurally and functionally interconnected, and their protein abundance was shown to be interdependent (*Maitland et al., 2019*). In agreement, *MAEA* deletion led to reduced amounts of RMND5A. Notably, RMND5A was stabilized in both wildtype and Y394A-mutated MAEA-expressing cells (*Figure 6I*, *Figure 6—source data 2*). We reasoned that association with MAEA, and presumably with RMND5A, is itself not sufficient to maintain UBE2H protein levels, but rather the catalytic activity of MAEA is required. A potential mechanism might be UBE2H autoubiquitylation and proteasomal degradation, which would be triggered in case ubiquitin-transfer-efficient UBE2H-CTLH modules are not formed. Proteome analysis by different studies identified four endogenous ubiquitylation sites on UBE2H (*Akimov et al., 2018*; *Kim et al., 2011*; *Wagner et al., 2011*; *Figure 6—figure supplement 1D*). In vitro ubiquitylation reaction with recombinant E1 and ubiquitin showed ubiquitylation of UBE2H but not of catalytic dead UBE2H-C87A mutant, indicating an autoubiquitylation mechanism (*Figure 6K*, *Figure 6—source data 2*). Overall, we conclude that UBE2H protein amounts are coupled to the presence of active MAEA and are controlled by differentiation-induced transcriptional regulation and post-transcriptionally by proteasomal degradation.

## Discussion

We demonstrate here that in-depth analysis of dynamic proteome profiles obtained from in vitro reconstituted erythropoiesis systems is an effective method to uncover differentiation stage-dependent expression of protein and protein complexes with functional roles in erythropoiesis. We observed that UBE2H and CTLH E3 complex assemblies form co-regulated E2-E3 modules required for erythropoiesis. Importantly, distinct protein profiles of the CTLH subunit homologues RANBP9 and RANBP10, suggest a remodelling of CTLH complex and the presence of erythroid maturation stage-dependent CTLH assemblies.

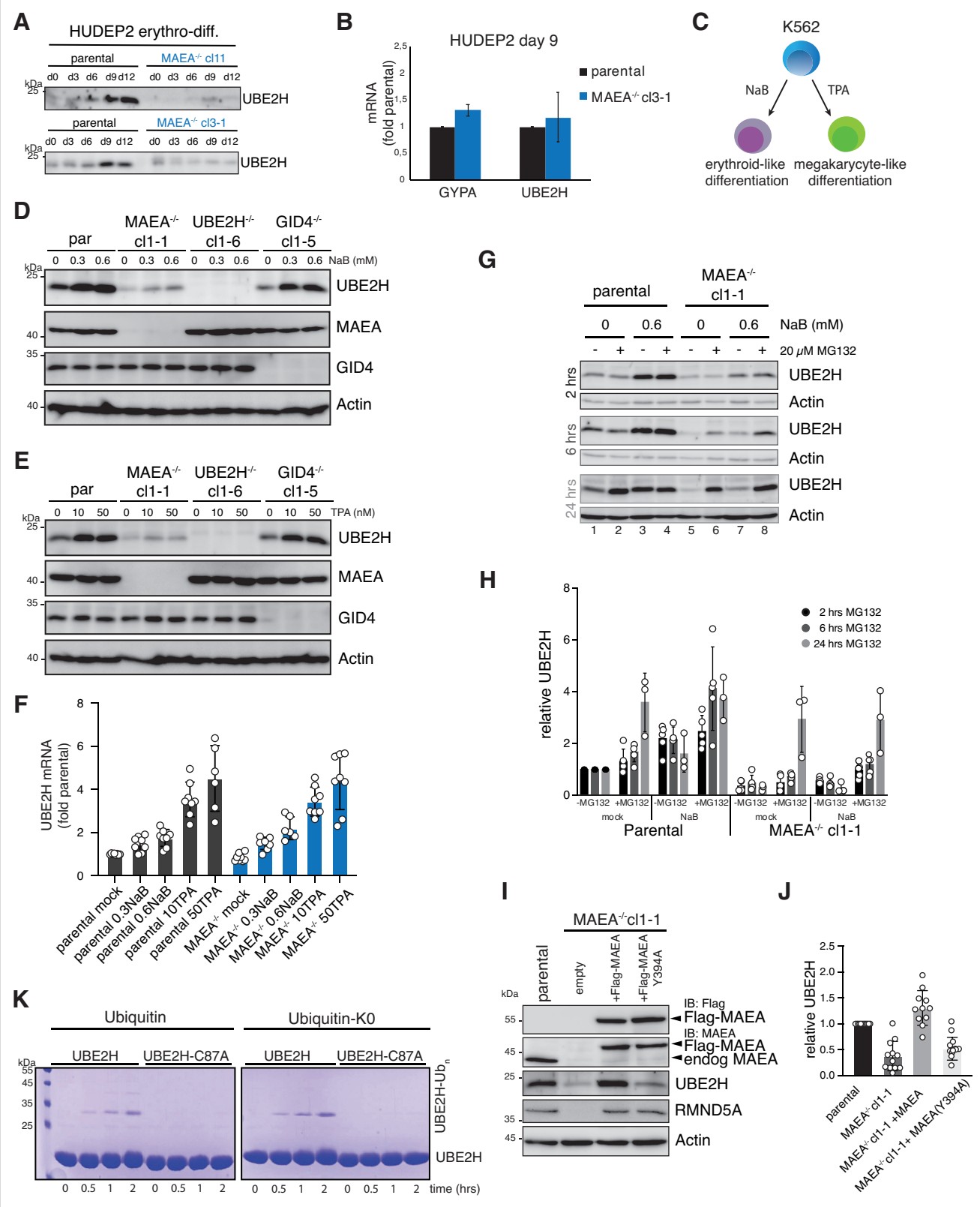

**Figure 6.** Cellular abundance of UBE2H is coupled to functional MAEA. (**A**) HUDEP2 parental and MAEA⁻/⁻ (clone 3-1 and 11) cells were differentiated in vitro and analysed by immunoblotting to detect UBE2H protein levels. (**B**) mRNA determination by RT-qPCR of HUDEP2 parental and MAEA⁻/⁻cl3-1 cells at differentiation stage day 9. Results (normalized to GAPDH) are mean ± SD of n = 2 experiments. (**C**) K562 cell can be either induced with Na-butyrate (NaB) for erythroid-like differentiation, or induced with 12-O-tetradodecanoyl-phorbol-13 acetate (TPA) for megakaryocyte-like differentiation.

*Figure 6 continued on next page*

*Figure 6 continued*

(**D**) K562 parental and knockout cell lines were treated with NaB for 24 hr and analysed by immunoblotting with indicated antibodies. (**E**) K562 parental and knockout cell lines were treated with TPA for 24 hr and analysed by immunoblotting with indicated antibodies. (**F**) K562 parental and MAEA$^{-/-}$cl1-1 cell lines were treated with either NaB or TPA for 24 hr, and UBE2H mRNA levels determination by RT-qPCR. Results (normalized to GAPDH) are mean ± SD of n = 6–8 experiments. (**G**) K562 parental and MAEA$^{-/-}$cl1-1 cell lines were mock- or 0.6 mM NaB-treated for 24 hr, followed by 2, 6, and 24 hr proteasome inhibition (MG132) and immunoblot analysis of cell lysates for UBE2H. Actin serves as protein loading control. (**H**) Quantitation of UBE2H immunoblot signals from (**G**) normalized to Actin and relative to parental mock values. Graph shows results by mean ± SD of n = 3–5 experiments. (**I**) Cell lysates of K562 parental and MAEA$^{-/-}$cl1-1 cells stably expressing Flag-MAEA wildtype (WT) or mutated Flag-MAEA-Y394A (Y394A) were analysed for UBE2H protein levels by immunoblotting. (**J**) Quantitation of immunoblots in (**I**) normalized to Actin and relative to parental values. Graph shows results by mean ± SD of n = 10–12 experiments. (**K**) Coomassie-stained SDS-PAGE gels presenting time course of in vitro autoubiquitylation assay with either ubiquitin or lysine-less ubiquitin (ubiquitin-K0) comparing UBE2H with UBE2H-C87A.

The online version of this article includes the following source data and figure supplement(s) for figure 6:

**Source data 1.** Original, uncropped scans of immunoblots.

**Source data 2.** Original, uncropped scans of immunoblots and Coomassie-stained SDS-PAGEs.

**Figure supplement 1.** MAEA-dependent UBE2H amounts in K562 cells.

**Figure supplement 1—source data 1.** Original, uncropped scans of immunoblots.

The activity and substrate specificity of multi-subunit E3 ligases are generally regulated by the control of complex subunit assembly. One of the best-studied E3 ligase complexes are members of the CRL family which engage interchangeable substrate receptor/adapter complexes for substrate-specific ubiquitylation. Substrate receptor assembly is kept in a highly dynamic state (*Reitsma et al., 2017*; *Straube et al., 2017*), whereby specific expression of substrate receptors in response to external and internal cellular cues, as well as in a cell-type and tissue-specific way, allows the formation of CRLs for selective and efficient client substrate ubiquitylation (*Gupta et al., 2013*; *McGourty et al., 2016*; *Ravenscroft et al., 2013*). Recently, the activity of the multiprotein yeast GID (orthologue of human CTLH complex) was shown to be predominantly regulated by engaging interchangeable substrate receptors, which conceivably target distinct substrates for degradation (*Chen et al., 2017*; *Chrustowicz et al., 2021*; *Kong et al., 2021*; *Langlois et al., 2022*; *Liu and Pfirrmann, 2019*; *Melnykov et al., 2019*; *Qiao et al., 2020*; *Santt et al., 2008*; *Shin et al., 2021*). However, the regulation of human CTLH activity via substrate receptor assembly is not well understood. Here we show that protein levels of GID4 did not significantly change in the HUDEP2 differentiation system, suggesting that GID4 may not be the critical substrate receptor of CTLH complex during erythropoiesis. However, the observed upregulation of the CTLH subunit WDR26, in conjunction with the recent evidence that WDR26 can function as a substrate receptor module (*Mohamed et al., 2021*), promotes the idea of a potential regulatory role of WDR26. Moreover, other, not yet identified substrate receptors might also exist and assemble with CTLH complex in an erythroid maturation-dependent manner. Besides WDR26, other CTLH subunits were significantly up- or downregulated with correlating protein intensity profiles indicating remodelling of CTLH assemblies during erythropoiesis. In particular, the homologous scaffold module subunits RANBP9 and RANBP10 displayed inverse protein expression profiles which agree with recently described mRNA levels (*Zhen et al., 2020*). As a consequence, HUDEP2 cells undergo a dynamic remodelling of mixed RANBP9/RANBP10-CTLH complexes, from predominantly RANBP9-containing CTLH complexes at progenitor/early erythroblast stages to predominantly RANBP10-CTLH complexes at later maturation stages. Importantly, RANBP9- and RANBP10-CTLH can assemble independently in cells, and additional structural/biochemical characterization of recombinant complexes revealed overall similar topologies of catalytic and scaffold modules of RANBP9-CTLH or RANBP10-CTLH, exerting E3 ligase activity in vitro with UBE2H.

Despite the compelling evidence of distinct CTLH assemblies in differentiating erythroid cells, we can only speculate about the mechanism of assembly and remodelling of complex subunits. RANBP9 and RANBP10 are unlikely to exchange freely within the CTLH complexes given that both subunits form extended surface interactions with ARMC8 and TWA1 forming the core of the scaffold module (*Sherpa et al., 2021*). Hence, RANBP9 and/or RANBP10 are likely to assemble CTLH complexes de novo, dependent on their availability and expression profiles. Importantly, our data showing evidence for RANBP9-CTLH, RANBP9/RANBP10-CTLH, and RANBP10-CTLH complexes substantiate the notion that CTLH complexes exist in multiple compositions and architectures, thereby expanding the complexity of the CTLH E3 family (*Sherpa et al., 2021*).

To determine the biological role of all possible CTLH complexes in erythropoiesis, we focused on *MAEA* knockout cells. MAEA deficiency in K562 and HUDEP2 cells caused significantly lower protein amounts of RMND5A and UBE2H, suggesting the ubiquitin transfer activity of most – if not all – cognate UBE2H-CTLH modules was eliminated. Analysis of *MAEA*- and *UBE2H*-deficient HUDEP2 cells, which were cultured under expansion growing (non-differentiating) or differentiation-induced conditions, showed increased haemoglobinization and enrichment of erythroid marker proteins, resembling a spontaneously and accelerated differentiated cell population. Interestingly, accelerated differentiation was also observed in *RANBP9*[-/-] cells. Hence, these knockout cells might be more susceptible to signals that promote erythropoiesis. Therefore, we reasoned that MAEA – in particular when assembled in RANBP9-CTLH complex – and UBE2H are either required to maintain HUDEP2 cells in a dormant/quiescent progenitor stage and/or to maintain accurate timing of early erythroid maturation. The in vitro reconstitution of erythropoiesis by the HUDEP2 system does not fully recapitulate in vivo erythropoiesis. However, our findings are supported by *MAEA* knockout studies in mice. Conditional *MAEA* deletion in murine haematopoietic stem cells (HSCs) impaired HSC quiescence, leading to a lethal myeloproliferative syndrome (*Wei et al., 2021*). The authors proposed a mechanism whereby the absence of MAEA leads to a stabilization of several haematopoietic cytokine receptors causing prolonged intracellular signalling (*Wei et al., 2021*). A similar concept might apply to the observed phenotypes of HUDEP2 *MAEA*[-/-] cells. Whereas proteins involved in quiescence and dormancy are not overall changed in these cells (*Figure 4—figure supplement 2G*), several erythroid plasma membrane proteins are overrepresented, including increased TFRC (transferrin receptor 1) levels. Hence, these cells are potentially more responsive to extracellular ferritin, enabling increased heme and haemoglobin production. Furthermore, mouse studies, which either conditionally deleted *MAEA* in central macrophages of erythroblastic islands or in erythroid progenitors, have revealed abnormal erythroblast maturation in the bone marrow showing altered profiles with distinct accumulation of maturation stages (*Wei et al., 2019*). This phenotype is, in part, recapitulated in *MAEA-*, *RANBP9-*, and *UBE2H*-deficient HUDEP2 cells by an apparent accumulation of early maturation stages. In addition, we observed inefficient enucleation of differentiating CD34[+] cells that are depleted of either *MAEA* or *UBE2H,* suggesting a role of UBE2H-CTLH at orthochromatic/reticulocytes stage. To date, no UBE2H knockout mouse models and erythropoiesis studies are available; however, our data are in agreement with studies of *MAEA* null mice (*Soni et al., 2006*). *MAEA*[-/-] embryos, which lack *MAEA* in central macrophages and erythroid progenitors, accumulate nucleated erythrocytes in peripheral blood. Cumulatively, our in vitro studies support the notion that the activity of UBE2H-CTLH modules is required at different erythroid maturation stages.

The functions of E2-E3 ubiquitylation modules are typically considered to be regulated via the E3 enzyme. However, E3 ubiquitylation involves different E2s, which themselves can be regulated by multiple mechanisms (*Stewart et al., 2016*), such as modulation by transcriptional/translational control (*Mejía-García et al., 2015*; *Whitcomb et al., 2009*; *Ying et al., 2013*). Apart from transcriptional upregulation of UBE2H in mammalian erythropoiesis (*Lausen et al., 2010*; *Wefes et al., 1995*), its *Drosophila* orthologue Marie Kondo (Kdo) was shown to be translationally upregulated upon oocyte-to-embryo transition (*Zavortink et al., 2020*), suggesting that UBE2H levels are regulatory nodes in developmental processes of higher eukaryotes. In agreement with transcriptional upregulation of UBE2H mRNA in terminal erythroid maturation (*Lausen et al., 2010*), we observe a substantial increase in UBE2H protein at orthochromatic stages of differentiating HUDEP2 cells. Surprisingly, absence of MAEA caused reduced protein but not the mRNA encoding UBE2H, suggesting an MAEA-dependent post-transcriptional mode of regulation. A MAEA mutant that still can bind UBE2H, but is defective in E3 ligase activity, does not efficiently rescue UBE2H levels in *MAEA*[-/-] cells. Therefore, in the absence of a ubiquitin-transfer-proficient MAEA (aka active UBE2H-CTLH complex), UBE2H can undergo autoubiquitylation at multiple lysine residues, which might by potentially triggering proteasomal degradation regulate UBE2H levels. Alternatively, a mechanism might be in place sensing inactive UBE2H-CTLH complexes that mediate ubiquitin targeting of UBE2H by other interacting E3 ligases, such as TRIM28 (*Doyle et al., 2010*). Future studies are needed to further dissect the precise mechanism. To our knowledge, UBE2H-CTLH is the first E2-E3 module described whereby E2 amounts are coupled to the activity of the cognate E3.

Cumulatively, our work features a mechanism of developmentally regulated E2-E3 ubiquitylation modules, which couples remodelling of multiprotein E3 complexes with its cognate E2 availability.

This mechanism assures assembly of distinct erythroid maturation stage-dependent UBE2H-CTLH modules, required for the orderly progression of human erythropoiesis, thus establishing a paradigm for other E2-E3 modules involved in developmental processes.

## Materials and methods

### HUDEP2, CD34⁺, and K562 cell culture and manipulation

HUDEP2 cells were cultured as described (*Kurita et al., 2013*). Immature cells were expanded in StemSpan serum-free medium (SFEM; Stem Cell Technologies) supplemented with 50 ng/ml human stem cell factor (hSCF) (R&D, #7466-SC-500), 3 IU/ml erythropoietin (EPO) (R&D, #287-TC-500), 1 µM dexamethasone (Sigma, #D4902), and 1 µg/ml doxycycline (Sigma, #D3072). Cell densities were kept within $50 \times 10^3$–$0.8 \times 10^6$ cells/ml and media replaced every other day. To induce erythroid maturation, HUDEP2 cells were cultured for 3 days (phase 1) in differentiation medium composed of IMDM base medium (GIBCO) supplemented with 2% (v/v) FCS, 3% (v/v) human serum AB-type, 3 IU/ml EPO, 10 µg/ml insulin, 3 U/ml heparin, 1 mg/ml holo-transferrin, 50 ng/ml hSCF, and 1 µg/ml doxycycline, followed by 9 days (phase 2) in differentiation medium without hSCF. Erythroid differentiation and maturation were monitored by flow cytometry (LSRFortessa, BD Biosciences) using PE-conjugated anti-CD235a/GYPA (BD Biosciences, clHIR2, #555570), FITC-conjugated anti-CD49d (BD Biosciences, cl9F10, #304316), and APC-conjugated anti-Band3/SLC4A1 (gift from Xiulan An lab, New York Blood Centre) and analysed with FlowJo software.

The erythroleukemia cell line K562 was obtained from ATCC (CCL-243) and cultured in IMDM (Gibco) supplemented with 10% (v/v) FCS (Gibco) and antibiotics (100 U/ml penicillin, 0.1 mg/ml streptomycin, Gibco), and regularly checked for the absence of mycoplasma contamination. To induce erythroid-like differentiation, K562 cells were treated with 0.3 mM or 0.6 mM Na-butyrate (NaB, Millipore) for 24 hr, and megakaryocyte-like differentiation was induced with 10 nM or 50 nM TPA (Sigma) for 24 hr. Where indicated, cells were treated with 10 µM proteasome inhibitor MG132 (Sigma) for different time phases.

CD34⁺ HSPCs were mobilized from normal subjects by granulocyte colony-stimulating factor, collected by apheresis, and enriched by immunomagnetic bead selection using an autoMACS Pro Separator (Miltenyi Biotec), in accordance with the manufacturer's protocol. At least 95% purity was achieved, as assessed by flow cytometry using a PE-conjugated anti-human CD34 antibody (Miltenyi Biotec, clone AC136, #130-081-002). A three-phase culture protocol was used to promote erythroid differentiation and maturation (*Giani et al., 2016*). In phase 1 (days 0–7), cells were cultured at a density of $2 \times 10^5$ cells/ml in IMDM with 2% human AB plasma, 3% human AB serum, 1% penicillin/streptomycin, 3 IU/ml heparin, 10 µg/ml insulin, 200 µg/ml holo-transferrin, 1 IU EPO, 10 ng/ml SCF, and 1 ng/ml IL-3. In phase 2 (days 8–12), IL-3 was omitted from the medium. In phase 3 (days 12–18), cells were cultured at a density of $10^6$ /ml, with both IL-3 and SCF being omitted from the medium and the holo-transferrin concentration being increased to 1 mg/ml. To quantify erythroblast enucleation, $2.0 \times 10^5$ CD34⁺-derived erythroid cells were incubated with Hoechst 33342 for 20 min at 37°C, fixed with 0.05% glutaraldehyde, and permeabilized with 0.1% Triton X-100. Cells were stained with FITC mouse anti-human CD235a (BD Biosciences, clone GA-R2, #561017) and then analysed by flow cytometry.

Gene disruption by CRISPR/Cas9:Cas9-sgRNA RNPs were generated by incubating 5 µg of purified Cas9 (from the University of California, Berkeley) with sgRNAs (at a molar ratio of 1:2) in a total volume of 5 µl in HF-150 buffer at room temperature for 25 min. The RNP cocktail was mixed with $2 \times 10^5$ CD34⁺ cells in a total volume of 20 µl in T buffer, then the cells were electroporated under these conditions with three pulses of 1600 V for 10 ms each, using a Neon Transfection System 10 µl kit (Thermo Fisher Scientific, #MPK1096). After electroporation, the cells were transferred to the culture medium for further analysis.

### Plasmid preparation and mutagenesis

The cDNAs for MAEA and UBE2H, corresponding to the canonical UniProt sequences, were obtained from human cDNA library (Max Planck Institute of Biochemistry). 3xFlag- and 6xMyc-tagged constructs, using pcDNA5-FRT/TO as parental vector, were generated by classic recombinant cloning methods.

Mutant versions of MAEA and UBE2H were prepared by the QuikChange protocol (Stratagene). All coding sequences were verified by DNA sequencing.

## K562 cell transfections and generation of CRISPR-Cas9 knockout cell lines

K562 cells were transformed by electroporation with Nucleofector Kit V (Bioscience, Lonza) according to the manufacturer's protocol. Briefly, $1 \times 10^6$ cells were harvested, washed once with 1× PBS (at room temperature), resuspended in 100 µl Nucleofector solution, and mixed with 5 µg plasmid DNA. After electroporation, cells were recovered in 3 ml medium and cultured for 48 hr. For IP experiments, three electroporation reactions with $1 \times 10^6$ cells were done in parallel, transformed cells pooled, and cultured for 48 hr.

$MAEA^{-/-}$, $MKLN1^{-/-}$, and $WDR26^{-/-}$ knockout cell lines were described previously (*Sherpa et al., 2021*). To generate CRISPR-Cas9-(D10A) nickase-mediated functional knockouts of UBE2H, paired sense and antisense guide RNAs (gRNA) were designed to target exon 2 of UBE2H (*Figure 4—figure supplement 1B*). Sense and antisense gRNAs were cloned into pBABED-U6-Puromycine plasmid (gift from Thomas Macartney, University of Dundee, UK) and pX335-Cas9(D10a) (Addgene) (*Cong et al., 2013*), respectively. The plasmid pair was co-transfected into K562 cells using Lipofectamine LTX reagent (Invitrogen) according to the manufacturer's protocol. Then, 24 hr after transfection, cells were selected in 2 µg/ml puromycin for 2 days, followed by expansion and single-cell dilution to obtain cell clones. Successful knockout of UBE2H was validated by immunoblot analysis and genomic sequencing of the targeted locus.

## Generation of CRISPR-Cas9-edited HUDEP2 knockout cell lines

To generate CRISPR-Cas9-(D10A) nickase-mediated functional knockouts of UBE2H in HUDEP2 cells, the same gRNA pair as described for the K562 knockout cell line has been used. For the functional knockouts of MAEA, RANBP9, and RANBP10 paired sense and antisense gRNAs were designed to target exon 2 (MAEA), exon 1 (RANBP9), and exon 7 (RANBP10) (*Figure 2—figure supplement 3 , Figure 4—figure supplement 1A*). The plasmid pairs were co-electroporated into HUDEP2 cells using Nucleofector Kit CD34+ (Bioscience, Lonza) according to the manufacturer's protocol. Then, 24 hr after transfection, cells were selected in 2 µg/ml puromycin for 2 days, followed by expansion and single-cell dilution to obtain cell clones. Cell densities were kept below $0.6 \times 10^6$ cells/ml throughout the process. Successful knockouts were validated by immunoblot analysis and genomic sequencing of targeted loci.

## Cell lysate preparation, immunoblot analysis, fractionation by sucrose density gradient, and immunoprecipitation

To generate K562 and HUDEP2 cell lysates, cells were harvested by centrifugation at $360 \times g$, washed once with ice-cold 1× PBS, and resuspended in lysis buffer (40 mM HEPES pH 7.5, 120 mM NaCl, 1 mM EGTA, 0.5% NP40, 1 mM DTT, and cOmplete Protease Inhibitor Mix [Roche]), and incubated on ice for 10 min. Cells were homogenized by pushing them 10 times through a 23G syringe. The obtained lysates were cleared by centrifugation at $23,000 \times g$ for 30 min at 4°C, and protein concentration was determined by Micro BCA-Protein Assay (Thermo Scientific, # 23235).

For immunoblot analysis, lysates were denatured with SDS sample buffer, boiled at 95°C for 5 min, separated on SDS-PAGE, and proteins were visualized by immunoblotting using indicated primary antibodies: RMND5A (Santa Cruz), MAEA (R&D Systems), RANBP9 (Novus Biologicals), RANBP10 (Invitrogen, #PA5-110267), TWA1 (Thermo Fisher), ARMC8 (Santa Cruz), WDR26 (Bethyl Laboratories), MKLN1 (Santa Cruz), YPEL5 (Thermo Fisher), GID4 (described in *Sherpa et al., 2021*), CD235a/GYPA (Abcam), HBD (Cell Signaling), HBG1/2 (Cell Signaling), and Flag (Sigma). Antibodies that recognize UBE2H were generated by immunizing sheep with GST-UBE2H (full length). Blots were developed using Clarity Western ECL Substrate (Bio-Rad, #16640474) and imaged using Amersham Imager 600 (GE Lifesciences). For quantitation described in *Figure 6F, H and J*, immunoblots from at least three biological repetitions were scanned with an Amersham Biosciences Imager 600 (GE Healthcare) and quantified using ImageJ software.

For the sucrose gradient fractionation, 3 mg of total protein were loaded onto a continuous 5–40% sucrose gradient (weight/volume in 40 mM HEPES pH 7.5, 120 mM NaCl, 1 mM EGTA, 0.5% NP40,

1 mM DTT, and cOmplete Protease Inhibitor Mix [Roche]) and centrifuged in a SW60 rotor at 34,300 rpm for 16 hr at 4°C. Thirteen 300 µl fractions were collected from top of the gradient, separated by SDS-PAGE, followed by immunoblotting using indicated antibodies.

Flag-tagged proteins were captured from 1 mg total cell lysate using anti-Flag affinity matrix (Sigma) for 1 hr at 4°C. For immunoprecipitation of endogenous proteins, 50 µg of antibody (UBE2H), 20 µg of nanobody (ARMC8), and 3.5 µg of antibody (RANBP9) were incubated overnight with 4 mg of cell lysate at 4°C. In all, 30 µl of Protein-G agarose (Sigma) was added and incubated for a further 2 hr. All immunoprecipitation reactions were washed in lysis buffer to remove nonspecific binding, immunoadsorbed proteins eluted by boiling in reducing SDS sample buffer, separated by SDS-PAGE followed by immunoblotting using indicated antibodies.

## Nanobody production

### Phage display selections
Purified human RANBP9-CTLH complex was coated on 96-well MaxiSorp plates by adding 100 µl of 1 µM proteins and incubating overnight at 4°C. Five rounds of phage display selections were then performed following standard protocols (*Tonikian et al., 2007*). The phage-displayed nanobody library used was reported before (*Nilvebrant and Sidhu, 2018*). Individual phage with improved binding properties obtained from rounds 4 and 5 were identified by phage ELISA and subjected to DNA sequencing of the phagemids to obtain nanobody sequences. Phage ELISA with immobilized proteins were performed as described before (*Zhang et al., 2016*).

### Cloning and protein purification
The nanobody cDNA was cloned into a vector containing either a C-terminal His tag (used for cryo-EM experiments) or an N-terminal GST tag (used for in vivo pulldown assays). The nanobody expression was carried out using BL21 pRIL cells and was purified from *Escherichia coli* using either an Ni-NTA or a glutathione affinity chromatography, followed by SEC in the final buffer containing 25 mM HEPES pH 7.5, 200 mM NaCl and 1 mM DTT.

## RNA isolation and RT-qPCR
Cells were treated with indicated concentrations of sodium butyrate (NaB) and TPA, respectively. 10 × 10$^6$ cells were lysed in 1 ml Trizol (Thermo Scientific, #15596018). Then, 200 µl chloroform (Fisher-Chemical, C496017) was added and samples were vigorously mixed. For phase separation, samples were then centrifuged at 10,000 × *g* for 10 min at 4°C. Subsequently, 400 µl of the upper clear phase were transferred into a new tube containing 500 µl isopropanol, mixed and incubated for 30 min on ice, followed by centrifugation at 10,000 × *g* at 4°C for 10 min. Pellet was washed once with 500 µl 70% ethanol and resuspended in RNase-free water (Invitrogen, 10977-035). Samples were stored at –80°C until analysis. cDNA was generated using SuperScript IV First Strand Synthesis System (Invitrogen, 18091050) according to the manufacturer's protocol. For qRT-PCR, cDNA, primers and SsoAdvanced Universal SYBR Green Supermix (Bio-Rad, 1725274) were mixed and run on a CFX96 Touch Deep Well Real Time PCR System (Bio-Rad) as per the manufacturer's protocol. The following forward/ reverse primer pairs were used: for GAPDH 5′ GTTCGACAGTCAGCCGCATC/5′ GGAATTTGCCATG GGTGGA; UBE2H 5′ CCTTCCTGCCTCAGTTATTGGC/5′ CCGTGGCGTATTTCTGGATGTAC; GYPA 5′ ATATGCAGCCACTCCTAGAGCTC/5′ CTGGTTCAGAGAAATGATGGGCA. Data was analysed with Bio-Rad CFX Manager using GAPDH for normalization.

## Protein expression and purification
The human RANBP10- and RANBP9-CTLH (including TWA1-ARMC8-MAEA-RMND5A with either RANBP10 or RANBP9) complexes were purified from insect cell lysates using StrepTactin affinity chromatography by pulling on the C-terminal twin Strep tag on TWA1, followed by anion-exchange chromatography and SEC in the final buffer containing 25 mM HEPES pH 7.5, 200 mM NaCl, and 5 mM (for cryo-EM) or 1 mM DTT (for biochemical assays). N-terminal GST-tagged version of hGID4(Δ1–99) (ΔGID4), wildtype UBE2H and UBE2H-C87A mutant were expressed in bacteria and purified by glutathione affinity chromatography followed by overnight cleavage using tobacco etch virus (TEV) protease. Further purification was carried out by anion-exchange chromatography followed by SEC in the final buffer containing 25 mM HEPES pH 7.5, 200 mM NaCl, and 1 mM DTT. To obtain saturated

RANBP10-CTLH complex with hGID4(Δ1–99) for cryo EM analysis, it was added in twofold excess to TWA1-ARMC8-MAEA-RMND5A-RANBP10 before final SEC.

Untagged WT ubiquitin used for in vitro assays was purified via glacial acetic acid method (*Kaiser et al., 2011*), followed by gravity S column ion-exchange chromatography and SEC.

## Ubiquitylation and autoubiquitylation assay

The in vitro multiturnover ubiquitylation assays with RANPB9-CTLH or RANBP10-CTLH complexes were performed using a C-terminally fluorescent-tagged model peptide with an N-terminal GID4-interacting sequence PGLW and a single lysine placed at the 23rd position from the N terminus. The reaction was started by mixing 0.2 µM Uba1, 1 µM UBE2H, 0.5 µM RANBP9-TWA1-ARMC8-RMND5A-MAEA or RANBP10-TWA1-ARMC8-RMND5A-MAEA complex, 1 µM GID4(Δ1–99), 1 µM fluorescent model peptide substrate, 20 µM Ub together with buffer containing ATP and MgCl$_2$. The reaction was quenched in sample loading buffer at different timepoints and visualized by scanning the SDS-PAGE in the Typhoon Imager (GE Healthcare).

The in vitro autoubiquitylation assay of the UBE2H was performed by incubating 30 µM of WT UBE2H or catalytically inactive UBE2H mutant (C87A UBE2H), 2 µM Uba1, 60 µM Ub or lysine-less Ub (K0-Ub) together with buffer containing ATP and MgCl$_2$. The reaction was performed at 30°C, and the reaction was quenched at 0, 30, 60, and 120 min by mixing the reaction with sample loading buffer. The assay was visualized by Coomassie-stained SDS-PAGE.

## Analytical SEC for RANBP10- and RANBP9-CTLH complexes

To see whether the RANBP10 and RANBP9 complexes assemble and migrate at similar molecular weight range, analytical sSEC was performed in a Superose 6 column (GE Healthcare) which was fitted to the Thermo Scientific Vanquish HPLC system. The column was equilibrated with 25 mM HEPES 7.5, 150 mM NaCl and 5 mM DTT, and 60 µl each of 10 µM purified RANBP10/TWA1/ARMC8/GID4(Δ1–99)/RMND5A/MAEA and RANBP9/TWA1/ARMC8/GID4(Δ1–99)/RMND5A/MAEA complexes were run through the HPLC system consecutively. The SEC fractions obtained were analysed with Coomassie-stained SDS-PAGE.

## Cryo-EM sample preparation and processing

Cryo-EM grids were prepared using Vitrobot Mark IV (Thermo Fisher Scientific) maintained at 4°C and 100% humidity. Then, 3.5 µl of the purified RANBP10-CTLH (RANBP10-TWA1-ARMC8-RMND5A-MAEA-GID4(Δ1–99)) complex at 0.6 mg/ml was applied to Quantifoil holey carbon grids (R1.2/1.3 300 mesh) that were glow-discharged separately in plasma cleaner. After sample application, grids were blotted with Whatman no. 1 filter paper (blot time: 3 s; blot force: 3) and vitrified by plunging into liquid ethane.

For the nanobody-bound RANBP9-CTLH complex, the purified nanobody was first mixed to the RANBP9-CTLH complex and ran on SEC. The peak fraction was then concentrated and prepared for cryo-EM using the same approach as mentioned above.

Both the cryo-EM data were collected on a Talos Arctica transmission electron microscope (Thermo Fisher Scientific) operated at 200 kV, equipped with a Falcon III (Thermo Fisher Scientific) direct electron detector, respectively. The data collection was carried out using EPU software (Thermo Fisher Scientific).

The cryo-EM data processing was carried out with Relion (*Fernandez-Leiro and Scheres, 2017*; *Scheres, 2012*; *Zivanov et al., 2018*). For processing the micrographs, frames were first motion-corrected using Relion's own implementation of MotionCor-like algorithm followed by contrast transfer function estimation using CTFFind 4.1. Particles were auto-picked using Gautomatch (http://www.mrc-lmb.cam.ac.uk/kzhang/) using a template of RANBP9-CTLH[SR4] (EMDB: EMD-12537). The extracted particles were subjected to several rounds of 2D classification and 3D classification followed by autorefinement without and with a mask. To improve the quality of maps obtained for RANBP10-CTLH complex, a focused 3D classification without particle alignment was performed with a mask over CTLH[SRS]. The best class with most features were chosen, and autorefinement with mask over the CTLH[SRS] was performed followed by post-processing.

## MS-based proteomics analysis of HUDEP2 samples

Cell pellets were lysed in SDC buffer (1% sodium deoxycholate in 100 mM Tris pH 8.5) and then heated for 5 min at 95°C. Lysates were cooled on ice and sonicated for 15 min at 4°C. Protein concentration

was determined by Tryptophan assay as described in *Kulak et al., 2014*, and equal amount of proteins were reduced and alkylated by 10 mM TCEP and 40 mM 2-chloroacetamide, respectively, for 5 min at 45°C. Proteins were subsequently digested by the addition of 1:100 LysC and Trypsin overnight at 37°C with agitation (1500 rpm). Next day, around 10 µg of protein material was processed using an in-StageTip (iST) protocol (*Kulak et al., 2014*). Briefly, samples were at least fourfold diluted with 1% trifluoro-acetic acid (TFA) in isopropanol to a final volume of 200 µl and loaded onto SDB-RPS Stag-eTips (Empore). Tips were then washed with 200 µl of 1% TFA in isopropanol and 200 µl 0.2% TFA/2% acetonitrile (ACN). Peptides were eluted with 80 µl of 1.25% ammonium hydroxide ($NH_4OH$)/80% ACN and dried using a SpeedVac centrifuge (Concentrator Plus; Eppendorf). MS loading buffer (0.2% TFA/2% ACN [v/v]) was added to the dried samples prior to LC-MS/MS analysis. Peptide concentrations were measured optically at 280 nm (Nanodrop 2000; Thermo Scientific) and subsequently equalized using MS loading buffer. Approximately 300–500 ng peptide from each sample was analysed using a 100 min gradient single-shot DIA method.

## LC-MS/MS analysis and data processing

Nanoflow LC-MS/MS measurements were carried out on an EASY-nLC 1200 system (Thermo Fisher Scientific) coupled to the Orbitrap instrument, namely, Q Exactive HF-X and a nano-electrospray ion source (Thermo Fisher Scientific). We used a 50 cm HPLC column (75 µm inner diameter, in-house packed into the tip with ReproSil-Pur C18-AQ1.9 µm resin [Dr. Maisch GmbH]). Column temperature was kept at 60°C with an in-house-developed oven.

Peptides were loaded in buffer A (0.1% formic acid [FA] [v/v]) and eluted with a linear 80 min gradient of 5–30% of buffer B (80% ACN and 0.1% FA [v/v]), followed by a 4 min increase to 60% of buffer B and a 4 min increase to 95% of buffer B, and a 4 min wash of 95% buffer B at a flow rate of 300 nl/min. Buffer B concentration was decreased to 4% in 4 min and stayed at 4% for 4 min. MS data were acquired using the MaxQuant Live software and a DIA mode (*Wichmann et al., 2019*). Full MS scans were acquired in the range of m/z 300–1650 at a resolution of 60,000 at m/z 200 and the automatic gain control (AGC) set to 3e6. Full MS events were followed by 33 MS/MS windows per cycle at a resolution of 30,000 at m/z 200 and ions were accumulated to reach an AGC target value of 3e6 or an Xcalibur-automated maximum injection time. The spectra were recorded in profile mode.

The single-shot DIA runs of HUDEP2 samples were searched with dDIA mode in Spectronaut version 14 (Biognosys AG) for final protein identification and quantification. All searches were performed against the human SwissProt reference proteome of canonical and isoform sequences with 42,431 entries downloaded in July 2019. Carbamidomethylation was set as fixed modification and acetylation of the protein N-terminus and oxidation of methionine as variable modifications. Trypsin/P proteolytic cleavage rule was used with a maximum of two miscleavages permitted and a peptide length of 7–52 amino acids. A protein and precursor FDR of 1% were used for filtering and subsequent reporting in samples (q-value mode).

## Bioinformatics data analysis

We mainly performed data analysis with Perseus (versions 1.6.0.9 and 1.6.1.3; *Tyanova et al., 2016*). Protein intensities were log2-transformed for further analysis. Data sets were filtered to make sure that identified proteins showed expression in all biological triplicates of at least one experimental group and the missing values were subsequently replaced by random numbers that were drawn from a normal distribution (width = 0.3 and down shift = 1.8). PCA of experimental groups and biological replicates was performed using Perseus. Multisample test (ANOVA) for determining whether any of the means of experimental group was significantly different from each other was applied to protein data set. For truncation, we used permutation-based FDR which was set to 0.01 or 0.05 in conjunction with an S0 parameter of 0.1. For hierarchical clustering of significant proteins, median protein abundances of biological replicates were z-scored and clustered using Euclidean as a distance measure for row and/or column clustering. GO enrichments in the clusters were calculated by Fisher's exact test using Benjamini–Hochberg FDR for truncation. Mean log2 ratios of biological triplicates and the corresponding p-values were visualized with volcano plots and significance was based on an FDR < 0.05 or 0.01. Network representation of significantly regulated proteins was performed with the STRING app (1.5.1) in Cytoscape (3.7.2).

## Acknowledgements

This work was supported by the Max Planck Society for the Advancement of Science and by the Deutsche Forschungsgemeinschaft (DFG, German Research Foundation) – SCHU 3196/1-1. We thank all the members of the department of Molecular Machines and Signaling at Max Planck Institute of Biochemistry for their assistance and helpful discussions, especially J Rajan Prabu for guidance in structural analysis, Susanne von Gronau for maintaining the insect cells, and Josef Kellermann for maintaining the lab. Xiuli An (Laboratory of Membrane Biology, New York Blood Center) provided the anti-Band3 antibody. We also thank Daniel Bollschweiler and Tillman Schäfer for maintaining the MPIB Cryo EM facility; Stephan Uebel and Stefan Pettera in the MPIB biochemistry core facility for the peptide synthesis.

## Additional information

### Competing interests

Brenda A Schulman: B.A.S. is adjunct faculty at St. Jude Children's Research Hospital, honorary professor at Technical University of Munich, a member of the scientific advisory bards of Interline Therapeutics and BioTheryX, and a convector of intellectual property related to DCN1 inhibitors licensed to Cinsano. The other authors declare that no competing interests exist.

### Funding

| Funder | Grant reference number | Author |
| --- | --- | --- |
| Deutsche Forschungsgemeinschaft | SCHU3196/1-1 | Brenda A Schulman |
| Max-Planck-Gesellschaft | | Brenda A Schulman |

The funders had no role in study design, data collection and interpretation, or the decision to submit the work for publication.

### Author contributions

Dawafuti Sherpa, Yu Yao, Data curation, Investigation; Judith Mueller, Peng Xu, Jakub Chrustowicz, Karthik V Gottemukkala, Christine Baumann, Annette Gross, Oliver Czarnecki, Data curation; Özge Karayel, Data curation, Formal analysis, Writing – review and editing; Wei Zhang, Jun Gu, Johan Nilvebrant, Sachdev S Sidhu, Resources; Peter J Murray, Conceptualization, Supervision, Writing – review and editing; Matthias Mann, Supervision, Writing – review and editing; Mitchell J Weiss, Conceptualization, Writing – review and editing; Brenda A Schulman, Conceptualization, Funding acquisition, Writing – review and editing; Arno F Alpi, Conceptualization, Data curation, Supervision, Writing - original draft, Writing – review and editing

### Author ORCIDs

Peng Xu http://orcid.org/0000-0002-7573-4813
Oliver Czarnecki http://orcid.org/0000-0002-1767-658X
Johan Nilvebrant http://orcid.org/0000-0002-6104-6446
Sachdev S Sidhu http://orcid.org/0000-0001-7755-5918
Peter J Murray http://orcid.org/0000-0001-6329-9802
Matthias Mann http://orcid.org/0000-0003-1292-4799
Mitchell J Weiss http://orcid.org/0000-0003-2460-3036
Brenda A Schulman http://orcid.org/0000-0002-3083-1126
Arno F Alpi http://orcid.org/0000-0002-9572-7266

### Decision letter and Author response

Decision letter https://doi.org/10.7554/eLife.77937.sa1
Author response https://doi.org/10.7554/eLife.77937.sa2

# Additional files

## Supplementary files
• Transparent reporting form

## Data availability
The mass spectrometry proteomics data have been deposited to the ProteomeXchange Consortium via the PRIDE partner repository with the dataset identifier PXD031992. The accession codes for the EM maps are available in EMDB as follows RANBP10-CTLH complex EMD-16242, ARMC8-specific nanobody bound CTLH complex EMD-16243.

The following previously published dataset was used:

| Author(s) | Year | Dataset title | Dataset URL | Database and Identifier |
|---|---|---|---|---|
| Xu P, Bludau I, Velan Bhoopalan S, Yao Y, Ana Rita FC, Santos A, Schulman BA, Alpi AF, Weiss MJ, Mann M, Karayel Ö | 2020 | Proteomic dissection of human erythropoiesis reveals critical roles of kinases | https://www.ebi.ac. uk/pride/archive/ projects/PXD017276 | PRIDE, PXD017276 |
| | | | , | |

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
