## [Editor Report]

This paper will be of interest to scientists in the field of hematology and ubiquitin biology. The work identifies previously unrecognized functions of and regulatory mechanisms impinging on CTLH E3 ubiquitin ligases during erythrocyte progenitor maintenance and differentiation. It provides new insights into the dynamic formation of E3 ubiquitin ligases during development, suggesting that rather than simply exchanging substrate adaptors, scaffolding proteins and collaborating E2 enzymes are also tightly regulated. The experiments are of high quality and a wealth of data supports the conclusions.

---

## [Decision Letter]

**Decision letter after peer review:**

Thank you for submitting your article "Differential UBE2H-CTLH E2-E3 ubiquitylation modules regulate erythroid maturation" for consideration by *eLife*. Your article has been reviewed by 3 peer reviewers, one of whom is a member of our Board of Reviewing Editors, and the evaluation has been overseen by a Reviewing Editor and Jonathan Cooper as the Senior Editor. The reviewers have opted to remain anonymous.

Essential revisions:

1. While the title of this study states that "differential UBE2H-CTLH E2-E3 ubiquitylation modules regulate erythroid maturation", the authors do not present data that support this conclusion. They show that RanBP9 and RanBP10 are differentially regulated in a manner that correlates with erythroid maturation, but they fall short of demonstrating that this change in expression actually results in differential complexes and regulates erythroid maturation. The authors therefore need to quantify the relative cellular amounts of RANBP9 and RANBP10 that are integrated into CTHL E3 at each differentiation state and correlate these to the relative changes in protein levels as determined by mass spec. In fact, they need to show that RANBP9 and RANBP10 form independent complexes in cells (this could be done by sequential IPs). Most importantly, they need to assess whether deletion of either RanBP9 or RanBP10 has functional consequences onto both the erythrocyte proteome as well as erythrocyte differentiation, which would indicate that the observations made here are significant in the context of this differentiation program.

2. In addition, the described phenotypes of UBE2H and MAEA deletion on erythrocyte differentiation need to be analyzed in more detail, in particular addressing whether the apparently accelerated differentiation is yielding functional progeny. While their data (in particular Figure 5 H) supports an accelerated differentiation phenotype, the authors should clarify whether MAEA- and UBE2H-deficient HUDEP2 cells yield functional progenitors. Could the authors perform assays to compare viability and functionality of control and CTLH E3 ligase-deficient orthochromatic erythroblasts (day 12 of differentiation)? Results from such analysis would help understanding how erythrocyte lineage-specific Maea knock out results in reduced circulating RBC counts (PMID: 30674470) (i.e. relative contributions of early versus late defects in erythropoiesis). Furthermore, is there any data from the UBE2H and/or MAEA KO proteomes compared to WT that shows decreased levels of proteins involved in maintaining a dormant/quiescent progenitor stage? The authors should analyze their datasets and address the possibility that they introduced in the discussion.

3. Finally, it is not clear how MAEA stabilizes UBE2H and thereby contributes to erythrocyte differentiation, and this part of the paper is therefore fairly observational without much insight into regulation or function. The authors show that re-expression of MAEA bearing a mutation in a residue they previously showed prevents ubiquitylation in vitro is not able to restore UBE2H expression, implying that the CTLH complex activity is required to maintain UBE2H expression. However, this claim is supported by one single transfection experiment, not quantified, which shows a slight reduction in UBE2H expression compared to wild-type MAEA. This is substandard. This experiment needs to be repeated and quantified. In addition, as it is known from both yeast and mammalian cell studies that MAEA and RMND5A expression are interdependent and that RMND5A expression is decreased upon MAEA knockout, it would be necessary to ensure that RMND5A expression is restored when re-expressing MAEA under the conditions used. To provide mechanistic insight, the authors might also want to test whether UBE2H levels are regulated through ubiquitin modification itself, as the activity of other E2s has been shown to be regulated by ubiquitylation (e.g. UBE2T (PMID: 16916645) or UBE2E (PMID: 25960396)). Finally, the effect of MG132 onto UBE2H levels could either be due to stabilization or effects on differentiation – other differentiation markers would have to be investigated. If these other markers also increase, the authors would need to formulate the text more carefully.

*Reviewer #1 (Recommendations for the authors):*

Suggestions for further improvement of the manuscript:

1. Figure 2: the most direct way to show that Ranbp9 and Ranbp10 assemble into independent complexes would be a sequential IP using either Ranbp9 or Ranbp10 and another CTLH complex subunit. If the authors are correct, then Ranbp9 complexes should not contain Ranbp10 and vice versa.

2. Lower resolution of Ranbp10 CTLH complexes and lower in vitro activity might indicate that some components or modifications are missing; these might provide additional regulation. It would have been nice if endogenous Ranbp9 and Ranbp10 IPs would have been analyzed by mass spectrometry.

3. Figure 6: effect of MG132 onto UBE2H levels could either be due to stabilization or effects on differentiation – other differentiation markers would have to be investigated. If these other markers also increase, I would formulate the text more carefully.

*Reviewer #2 (Recommendations for the authors):*

1. The authors overstate some of their conclusions. The title states that "differential UBE2H-CTLH E2-E3 ubiquitylation modules regulate erythroid maturation". The authors do not present data that support this conclusion. They show that MAEA and UBE2H knockouts impair erythroid maturation, consistent with CTLH complex involvement in this process. And they show that RanBP9 and RanBP10 are differentially regulated in a manner that correlates with erythroid maturation, but they fall short of demonstrating that this change in expression, which would result in differential complexes actually results in or regulates erythroid maturation.

2. Another conclusion that is overstated, is that RanBP9 and RanBP10 assemble in "distinct CTLH E3 supramolecular complexes" which would be of importance to support the conclusion of "structurally distinct complexes" regulating erythroid maturation. However, they did not provide evidence that RanBP9 and RanBP10 form different complexes in vivo. The authors show that RanBP9 and RanBP10 can assemble with ArmC8 and Twa1 to form CTLH complex core module in vitro, but the data from the sucrose gradients is in support of a model where RanBP9 and RanBP10 co-exist in the complex. Therefore, as stated by the authors in the discussion (line 492) "…we can only speculate about the mechanism of assembly of complex subunits", which makes it difficult to support their claim that "differential ubiquitylation modules regulate erythroid maturation".

3. Another main conclusion stated by the authors is that re-expression of MAEA bearing a mutation in a non-priming residue (Y394A, that they previously showed prevented ubiquitination in vitro) is not able to restore UBE2H expression, implying that the CTLH complex activity is required to maintain UBE2H expression. However, this claim is supported by one single transfection experiment, not quantified, which shows a slight reduction in UBE2H expression compared to wild-type MAEA. This is substandard. This experiment needs to be repeated and quantified. In addition, as it is known from both yeast and mammalian cell studies that MAEA and RMND5A expression are interdependent and that RMND5A expression is decreased upon MAEA knockout, it would be necessary to ensure that RMND5A expression is restored when re-expressing MAEA under the conditions used.

4. Adding to the above comment. This report convincingly implicates MAEA in erythropoiesis signaling, however, the connection with the CTLH complex reported by the authors is puzzling. The discussion introduces two other publications that connect MAEA and hematopoietic signaling, but the manuscript overall fails to make any other connections with the CTLH complex. What is particularly surprising is that deletion of WDR26 – a complex member that was previously shown to regulate supramolecular formation in the same cell line used in this study – is reported to have no effect on UBE2H levels compared to parental cells. The same goes for GID4, the receptor subunit of the complex, which appears to be dispensable for UBE2H regulation. Could MAEA be functioning independently of the CTLH complex to induce this phenotype? Do RMND5A knockout cells exhibit the same phenotype, since it is predicted that only the MAEA-RMND5A RING heterodimer is functional?

Other Comments

1. Line 197: Authors suggest that RanBP9 and RanBP10 protein levels change from day 0 to day 6 of differentiated HUDEP2 cells based off the sucrose gradient, but since there is no loading control in this type of experiment, they cannot use sucrose gradients to make this conclusion. Curiously, the authors do not see a difference in any other complex member protein levels by sucrose gradient, even though the authors conclude that there are protein level differences based on Figure 1F and G. A Western blot showing levels of complex members on day 0 and day 6, with appropriate loading controls, would allow for these conclusions while also validating the proteomic data shown in Figure 1. Also, why choose day 0 and day 6 for Figure 2 instead of day 0 and day 12? Day 12 showed the largest effects according to data in Figure 1.

2. Line 201: ARMC8 was immunoprecipitated using a nanobody, but in the methods section (line 651), authors indicate that ARMC8 antibody was used for immunoprecipitation. This ambiguity needs to be addressed. Authors should also indicate whether the nanobody is specific for ARMC8 α or β; since this nanobody has not been used in previous publication, ARMC8 KO cell lines should be the appropriate negative control instead of "mock".

3. Line 209: a couple of complex members besides RanBP9 and RanBP10 should be immunoblotted on the sucrose gradients, especially since the purpose of the figure was to demonstrate "whether RanBP9 and RanBP10 could can independently form CTLH complexes".

4. Line 214: authors mention that RanBP9 and RanBP10 assemble in "distinct CTLH E3 supramolecular complexes" based on Figure 2, but as seen in Figure 2A, both RanBP9 and RanBP10 exhibit a similar sedimentation pattern in parental HUDEP2 cells, suggesting that both proteins may be present in the same complex rather than RanBP9- and RanBP10-dependent complexes. Since this is a fundamental conclusion from this paper, authors need to further analyze how RanBP9 and RanBP10 interact with the complex when both are present. The authors mention (Line 495) a previous study that showed that other CTLH complex members' expression is compromised upon RanBP9 depletion, and they mention that RanBP9 is "slightly increased" in RanBP10 KO cells, although no quantifications are shown. Did the authors observe a similar decrease in CTLH complex subunits upon RanBP9 knockout as previously reported, or is there a compensatory effect in HUDEP2 cells? This would help better understand the nature of the complexes.

5. Figure 3: it has previously been shown that GID4 is larger than TWA1 (Lampert et al., 2019; Qiao et al., 2020; Mohamed et al., 2021), yet Figure 3A indicates the opposite.

6. Line 223: Sherpa et al., 2021 showed that when WDR26 is deleted, much of the CTLH complex population migrates to a lower molecular weight when performing sucrose gradients in vivo using K562 cells, yet the authors omitted WDR26 when reconstituting the complex in vitro. Given how supramolecular assembly, which corresponds to data shown in Figure 2A, is governed by WDR26 in human cells, there needs to be a discussion about why this protein was omitted in this experiment.

7. Line 478: authors mention that they have shown no differences in GID4 levels during erythropoiesis in HUDEP2 cells, but Figure 1F does not include GID4. Even though it may be in the supplemental dataset, it would be logical to include GID4 in Figure 1F as well.

8. Line 506: the authors state: "MAEA deficiency in K562 and HUDEP2 cells caused significantly lower protein amounts of RMND5a and UBE2H…”. There is no data, or citation to be found in the manuscript that shows RMND5a depletion in HUDEP2 cells.

9. A recent paper (Zhen et al., 2020) showed that CTLH complex member mRNA levels decrease as human erythroblasts become terminally differentiated, yet Figure 1G shows the opposite effect on CTLH protein level in the same system with the notable exceptions being RanBP9 and RanBP10. Could any of your results explain why this may be?

10. Line 512: is there any data from the UBE2H and/or MAEA KO proteomes compared to WT that shows decreased levels of proteins involved in maintaining a dormant/quiescent progenitor stage? The authors should analyze their datasets and address the possibility that they introduced in the discussion.

11. The model proposed in Figure 6J was not addressed in the manuscript. Is there altered hematopoiesis if RanBP9 or RanBP10 are knocked out? If the CTLH complex is going to be associated with this pathway, there needs to be evidence showing that deletion of another complex member leads to a similar phenotype.

*Reviewer #3 (Recommendations for the authors):*

To address the weaknesses I outlined above, I have the following comments and suggestions for experiments:

1) Differentiation stage-specific formation of RANBP9-CTHL and RANBP10-CTHL complexes:

– While there is good evidence that there is differential formation of these of RANBP9-CTHL and RANBP10-CTHL during maintenance and 6 days of erythrocyte progenitor differentiation (Figure 2A,B) , it would greatly strengthen if the authors could quantify the effects. E.g. would it be possible to quantify the relative cellular amounts of RANBP9 and RANBP10 integrated into CTHL E3 at each differentiation state (input versus fraction 9,10,11 from sucrose gradients) and correlate these to the relative changes in protein levels as determined by mass spec? Alternatively the endogenous ARMC8 IPs could be quantified in the same manner. In case there is a clear correlation between the relative protein level increase and the relative integration of RANBP9 and RANBP10, such data would provide evidence that changes in CTHL composition are driven by abundance of RANBP9/10 withing cells (as suggested by the authors in the discussion). Alternatively, if there is no strict correlation, this could point to cellular mechanisms that regulate their assembly into the CTHL complex.

– To provide evidence that these differentiation-specific RANBP9-CTHL and RANBP10-CTLH are functionally distinct and important for erythropoiesis, could the authors use their available RANBP9/10 knock out cells and determine effects on progenitor maintenance and differentiation?

2) Coupling of UBE2H stability to CTLH E3 ligase activity:

– The authors convincingly show that UBE2H stability depends on the catalytic activity of the CTLH E3 ligase and in its absence is targeted for proteasomal degradation. This is an conceptually interesting finding and it would greatly strengthen the manuscript if the authors could provide evidence for when and how such regulation is important during erythrocyte progenitor maintenance or differentiation. E.g. would overexpression of UBE2H dysregulate erythrocyte progenitor maintenance, in which overall CTHL E3 ligase function is presumably lower?

– Can the authors speculate on how catalytic activity of MAEA prevents proteasomal degradation of UBE2H? Could it be through ubiquitin modification itself? The activity of other E2s have been shown to be regulated by ubiquitylation (e.g. UBE2T (PMID: 16916645) or UBE2E (PMID: 25960396)), so it might be worthwhile to test whether UBE2H undergoes MAEA-dependent, regulatory ubiquitylation, which, if lost, leads to proteasomal degradation.

3) CTLH E3 ligase loss and accelerated erythrocyte differentiation: while the data presented (in particular Figure 5 H) supports an accelerated differentiation phenotype, I think the authors should clarify whether they think that MAEA- and UBE2H-deficient HUDEP2 cells yield functional progenitors. At least from the proteomic data analysis in Figure 4H and J one would expect that the resulting cell populations are quite different. Could the authors perform assays to compare viability and functionality of control and CTLH E3 ligase-deficient orthochromatic erythroblasts (day 12 of differentiation)? Results from such analysis would also help understanding how erythrocyte lineage-specific Maea knock out results in reduced circulating RBC counts (PMID: 30674470) (i.e. relative contributions of early versus late defects in erythropoiesis).

Additional comments:

Line 507: the authors state that MAEA deficiency results in lower RMND5a amounts. I hope I am not missing something, by I could not easily see the data for this in the figures. Could the authors please clarify and cite the appropriate figure showing this?

---

## [Author Response]

Essential revisions:1. While the title of this study states that "differential UBE2H-CTLH E2-E3 ubiquitylation modules regulate erythroid maturation", the authors do not present data that support this conclusion. They show that RanBP9 and RanBP10 are differentially regulated in a manner that correlates with erythroid maturation, but they fall short of demonstrating that this change in expression actually results in differential complexes and regulates erythroid maturation. The authors therefore need to quantify the relative cellular amounts of RANBP9 and RANBP10 that are integrated into CTHL E3 at each differentiation state and correlate these to the relative changes in protein levels as determined by mass spec. In fact, they need to show that RANBP9 and RANBP10 form independent complexes in cells (this could be done by sequential IPs). Most importantly, they need to assess whether deletion of either RanBP9 or RanBP10 has functional consequences onto both the erythrocyte proteome as well as erythrocyte differentiation, which would indicate that the observations made here are significant in the context of this differentiation program.

We provide in our revised manuscript a comprehensive set of new data to support and strengthen our conclusion that modular UBE2H-CTLH complexes regulate erythroid maturation.

A sequential IP approach (as suggested by the reviewers’) was established, allowing us to discriminate and monitor endogenous RANBP9- versus RANBP10-assembled CTLH complexes during erythroid differentiation (outlined Figure 2D). In the first step a RANBP9 antibody was used to precipitate all RANBP9-assembled CTLH complexes. In the second step, the RANBP9-depleted supernatant was subjected to an IP with our developed ARMC8 nanobody to capture remaining RANBP10-CTLH complexes (Despite testing several commercial RANBP10 antibodies, none of these were specific and efficient in IPs). We applied this approach to cell lysate samples derived from different erythroid differentiation stages (at day 0, 4, 8) and monitored relative levels of RANBP9- and RANBP10-assembled CTLH complexes by immunoblotting. In accordance with changes of RANBP9 and RANBP10 levels (as shown in the proteomics data), we observed a dynamic modulation of mixed RANBP9 and RANBP10-CTLH complexes, from predominantly RANBP9-CTLH complexes at progenitor/early erythroblast stage to predominantly RANBP10-CTLH complexes at later differentiation stages (Figure 2E). Furthermore, the sequential IPs performed provide evidence for the assembly of independent RANBP9-CTLH and RANBP10-CTLH complexes. These data – including the sedimentation experiments (Figure 2C and 2G) – suggest the presence of distinct stage-specific assemblies of RANBP9- and RANBP10-CTLH complexes in terminal erythropoiesis.

To assess whether these RANBP9- and RANBP10-CTLH complexes have a role in the differentiation programme we generated a set of CRISPR-Cas9-mediated *RANBP9^-/-^* and *RANBP10^-/-^* deletion cell lines. Following up on the spontaneous and accelerated differentiation phenotypes described for *UBE2H* and *MAEA* deletions, we first analysed *RANBP9^-/-^* and *RANBP10^-/-^* cells for altered expression of CD235a/GYPA. *RANBP9^-/-^*, but not *RANBP10^-/-^* cells (analysis of knock-out pools as well as single clones) showed increased expression of CD235a/GYPA (Figure 5A and 5D), and an increased population of CD235a/GYPA expressing cells when cultured under non-differentiating conditions (Figure 5B and 5E). Moreover, when differentiation was induced in these deletion lines, only *RANBP9^-/-^* clones showed higher CD49d^+^/Band3^+^ populations, indicating accelerated differentiation and mirroring *UBE2H* and *MAEA* deletions. We believe these new data support our conclusion that RANBP9- and RANBP10-CTLH complexes are required at different stages of differentiation, and that RANBP9-CTLH has a potential role in maintaining HUDEP2 cells in dormant/quiescent progenitor cells and/or controlling transition to terminal differentiation.

2. In addition, the described phenotypes of UBE2H and MAEA deletion on erythrocyte differentiation need to be analyzed in more detail, in particular addressing whether the apparently accelerated differentiation is yielding functional progeny. While their data (in particular Figure 5 H) supports an accelerated differentiation phenotype, the authors should clarify whether MAEA- and UBE2H-deficient HUDEP2 cells yield functional progenitors. Could the authors perform assays to compare viability and functionality of control and CTLH E3 ligase-deficient orthochromatic erythroblasts (day 12 of differentiation)? Results from such analysis would help understanding how erythrocyte lineage-specific Maea knock out results in reduced circulating RBC counts (PMID: 30674470) (i.e. relative contributions of early versus late defects in erythropoiesis). Furthermore, is there any data from the UBE2H and/or MAEA KO proteomes compared to WT that shows decreased levels of proteins involved in maintaining a dormant/quiescent progenitor stage? The authors should analyze their datasets and address the possibility that they introduced in the discussion.

Our initial in vitro erythropoiesis studies with HUDEP2 cells show spontaneous and accelerated erythroid maturation in MAEA and UBE2H deficient cells. Importantly mouse studies, that either conditionally deleted *MAEA* in central macrophages of erythroblastic islands or in erythroid progenitors (Wei et al., 2019; PMID: 30674470), have revealed abnormal erythroblast maturation in the bone marrow showing altered profiles with distinct accumulation of maturation stages. This phenotype is, at least in part, recapitulated in our study of *MAEA-* and *UBE2H-*deficient HUDEP2 cells by an apparent accumulation of early maturation stages.

We followed up on these data and addressed reviewer’s comment regarding the functionality of *MAEA-* and *UBE2H*- deficient orthochromatic erythroblasts that might account for reduced circulating RBC counts in erythrocyte lineage-specific *MAEA* knockout (Wei et al., 2019; PMID: 30674470), or anaemia in *MAEA* null mice embryos, which also accumulate nucleated erythrocytes in peripheral blood (Soni S. et al. 2006, PMID: 16707498). The maturation of orthochromatic erythroblasts into reticulocytes is characterized by the elimination of the condensed nucleus – enucleation. Hence, we focused our functional analysis of *MAEA-* and *UBE2H*- deficient orthochromatic erythroblasts by monitoring enucleation efficiency as a critical step in generating functional reticulocytes. HUDEP2 cells show intrinsically weak enucleation efficiency, we therefore switched to the CD34^+^ erythroid maturation system. We observed inefficient enucleation of maturating CD34^+^ cells that were depleted of either *MAEA* or *UBE2H,* suggesting a role of UBE2H-CTLH at ortho/reticulocytes stage (Figure 4I-4L). To date, no UBE2H knock out mouse models and erythropoiesis studies are available, however, our data agree with data from *MAEA* null mice that accumulate nucleated erythrocytes in peripheral blood (Soni S. et al. 2006, PMID: 16707498). Cumulatively, our in vitro studies support the notion that the UBE2H-CTLH modules are required in erythroid maturation.

Reviewers raised an interesting point about whether proteins involved in maintaining dormant/question progenitor stage might be altered in *MAEA* and *UBE2H* deletion cells. To address this question, we provide a graph with protein fold change *UBE2H^-/-^* vs parental plotted against protein fold change *MAEA^-/-^* vs parental and highlighted proteins that have been implicated in quiescence/dormancy including GATA2 (Figure 4—figure supplement 2G). However, none of these are significantly changing in *MAEA* and *UBE2H* deletion cells and are therefore unlikely the cause of the spontaneous and accelerated maturation phenotype. The point is mentioned in the discussion of the revised manuscript.

3. Finally, it is not clear how MAEA stabilizes UBE2H and thereby contributes to erythrocyte differentiation, and this part of the paper is therefore fairly observational without much insight into regulation or function. The authors show that re-expression of MAEA bearing a mutation in a residue they previously showed prevents ubiquitylation in vitro is not able to restore UBE2H expression, implying that the CTLH complex activity is required to maintain UBE2H expression. However, this claim is supported by one single transfection experiment, not quantified, which shows a slight reduction in UBE2H expression compared to wild-type MAEA. This is substandard. This experiment needs to be repeated and quantified. In addition, as it is known from both yeast and mammalian cell studies that MAEA and RMND5A expression are interdependent and that RMND5A expression is decreased upon MAEA knockout, it would be necessary to ensure that RMND5A expression is restored when re-expressing MAEA under the conditions used. To provide mechanistic insight, the authors might also want to test whether UBE2H levels are regulated through ubiquitin modification itself, as the activity of other E2s has been shown to be regulated by ubiquitylation (e.g. UBE2T (PMID: 16916645) or UBE2E (PMID: 25960396)). Finally, the effect of MG132 onto UBE2H levels could either be due to stabilization or effects on differentiation – other differentiation markers would have to be investigated. If these other markers also increase, the authors would need to formulate the text more carefully.

We observed in *MAEA* knock-out cell lines (HUDEP2 and K562) reduced levels of UBE2H (proteomics data, immunoblot analysis), which can be rescued by expressing wildtype MAEA, but not a catalytic defective MAEA-Y394A mutant. To strengthen our conclusion that MAEA activity is required for UBE2H levels, we generated clonal MAEA^-/-^ cell lines stably expressing either wildtype or MAEA-Y394A mutant (note, clones were selected for comparable expression levels of wildtype or MAEA-Y394A). Immunoblot analysis in revised Figure 6I confirms that MAEA-Y394A is not efficient in stabilizing UBE2H, whereas wild type completely rescued UBE2H levels. Immunoblot signals of UBE2H levels (n>10), normalized to actin have been quantified indicating statistical significance (Figure 6J). Importantly, RMND5A levels are stabilized by wildtype MAEA expression, as well as MAEA-Y394A.

We agree with the reviewer’s point that the effect of MG132 onto UBE2H levels could be either due to stabilization or via effects on differentiation. To investigate both effects, the original experiment was designed and carried out in the absence/presence of the erythroid differentiation inducer NaB and MG132. Whereas NaB treatment only had a modest effect on the stabilization of UBE2H in MAEA^-/-^ cells (Figure 6G, compare lane 5 with 7), prolonged MG132 treatment resulted to UBE2H levels as observed in parental cells (Figure 6G compare lane 5 with 6). In fact, MG132 treatment reduced rather increased levels of the erythroid maker GYPA. Hence, rescue of UBE2H levels is unlikely due to differentiation upon MG132 treatment.

To follow up on reviewers’ comments and suggestions regarding a potential mechanism regulating UBE2H levels in the absence of functional MAEA, we investigated whether “orphan” UBE2H might undergo autoubiquitylation. Proteomics studies identified several UBE2H ubiquitylation sites (RMIDs:29967540, 2106985, 21890473), which we highlighted in an Alphafold 2-derived model structure of UBE2H-Ub (Figure 6—figure supplement 1K). The surface-exposed sites that are in close proximity to the catalytic C87 could be potentially autoubiquitylated. We reconstituted autoubiquitylation in an in vitro assay and detected ubiquitylation of UBE2H which was dependent on the catalytic C86 residue (Figure 6L). Moreover, using lysine-less ubiquitin with all Lys-to-Arg mutation, indicated predominant UBE2H monoubiquitylation. Taken together, we propose a plausible mechanism of MAEA-dependent UBE2H levels: it is likely that in the absence of ubiquitin-transfer proficient MAEA (aka functional CTLH E3 ligase complex), UBE2H undergoes autoubiquitylation and subsequent proteasomal degradation. Alternatively, other UBE2H-interacting E3 ligases, such as TRIM28 (Doyle et al. 2010, PMID: 20864041), could potentially target UBE2H for proteasomal degradation.

Reviewer #1 (Recommendations for the authors):Suggestions for further improvement of the manuscript:1. Figure 2: the most direct way to show that Ranbp9 and Ranbp10 assemble into independent complexes would be a sequential IP using either Ranbp9 or Ranbp10 and another CTLH complex subunit. If the authors are correct, then Ranbp9 complexes should not contain Ranbp10 and vice versa.

We followed reviewer’ s suggestion and established a sequential IP, that allowed us to discriminate and monitor endogenous RANBP9- versus RANBP10-assembled CTLH complexes during erythroid differentiation (outlined in Figure 2D). In the first step a RANBP9 antibody was used to precipitate all RANBP9-assembled CTLH complexes. In the second step, the RANBP9-depleted supernatant was subjected to an IP with our developed ARMC8 nanobody to capture the remaining RANBP10-CTLH complexes. (Several commercial RANBP10 antibodies were tested, however none of these were specific and efficient in IPs). We applied this approach to cell lysates derived from different erythroid maturation stages (at day 0, 4, 8) and monitored relative amounts of RANBP9- and RANBP10-assembled CTLH complexes by immunoblotting. In accordance with changes of RANBP9 and RANBP10 levels (as shown in the proteomics data), we observed a dynamic modulation of mixed RANBP9 and RANBP10-CTLH complexes, from predominantly RANBP9-CTLH complexes at progenitor/early erythroblast stage to predominantly RANBP10-CTLH complexes at later differentiation stages (Figure 2E). The sequential IPs performed provide further evidence for the assembly of independent RANBP10-CTLH complexes (Figure 2E, Pellet 2).

Moreover, upon depleting RANBP9 or RANBP10, we can show that RANBP9- or RANBP10-CTLH supramolecular assemblies can independently assemble (Figure 2G). Taken together, we suggest that in cells such mixed population of RANBP9- and RANBP10-CTLH complexes exist, with preference dependent on the levels of RANBP9 and RANBP10 during differentiation stages.

2. Lower resolution of Ranbp10 CTLH complexes and lower in vitro activity might indicate that some components or modifications are missing; these might provide additional regulation. It would have been nice if endogenous Ranbp9 and Ranbp10 IPs would have been analyzed by mass spectrometry.

Regarding the low resolution of RANBP10-CTLH, we only performed a short data collection in Talos Arctica transmission electron microscope (Thermo-Fisher Scientific) operated at 200 kV. The goal was not meant to achieve high-resolution map but rather to obtain a modest resolution RANBP10-CTLH map that allowed us to compare the overall structural topology with the previously resolved RANBP9-CTLH map (EMD:12537).

Despite the overall similar topology of RANBP9- and RANBP10-CTLH complexes, subtle differences in model substrate-peptide binding and/or orientation of lysine targets might account for the lower ubiquitin-transfer activity observed in this particulate experimental in vitro assay system. Moreover, the N-termini of RANBP9 and RANBP10 are significantly different (Figure 2—figure supplement 1A), which could allosterically influence complex activity or substrate targeting.

We agree that RANBP9 and RANBP10 complexes might also assemble additional subunits and this is an exciting question for our future studies. However, IP/MS experiments and follow-up analysis of interaction partners would exceed the scope of the manuscript.

3. Figure 6: effect of MG132 onto UBE2H levels could either be due to stabilization or effects on differentiation – other differentiation markers would have to be investigated. If these other markers also increase, I would formulate the text more carefully.

We agree with reviewers’ point that the effect of MG132 onto UBE2H levels could be either due to stabilization or via effects on differentiation. To investigate both effects, the original experiment was designed and carried out in the absence/presence of the erythroid differentiation inducer NaB and proteasome inhibitor MG132. Whereas NaB treatment only had a modest effect on the stabilization of UBE2H in MAEA^-/-^ cells (Figure 6G compare line 5 with 7), prolonged MG132 treatment resulted to UBE2H levels as observed in parental cells (Figure 6G**,** compare line 5 with 6). In fact, MG132 treatment reduced rather increased levels of the erythroid maker GYPA. Hence, the rescue of UBE2H levels is unlikely due to differentiation upon MG132 treatment.

Reviewer #2 (Recommendations for the authors):1. The authors overstate some of their conclusions. The title states that "differential UBE2H-CTLH E2-E3 ubiquitylation modules regulate erythroid maturation". The authors do not present data that support this conclusion. They show that MAEA and UBE2H knockouts impair erythroid maturation, consistent with CTLH complex involvement in this process. And they show that RanBP9 and RanBP10 are differentially regulated in a manner that correlates with erythroid maturation, but they fall short of demonstrating that this change in expression, which would result in differential complexes actually results in or regulates erythroid maturation.

To assess whether these RANBP9- and RANBP10-CTLH complexes have a role in the erythroid maturation programme we generated set of CRISPR-Cas9-mediated *RANBP9^-/-^* and *RANBP10^-/-^* deletion cell lines. Following up on the spontaneous and accelerated differentiation phenotypes described for *UBE2H* and *MAEA* deletions, we first analysed *RANBP9^-/-^* and *RANBP10^-/-^* cells for altered expression of CD235a/GYPA. By analysing knock-out pools as well as single clones, we observed that *RANBP9^-/-^* but not *RANBP10^-/-^* cells have increased expression of CD235a/GYPA by immunoblot (Figure 5A and 5D) and flow cytometry revealed an increased population of CD235a/GYPA expressing cells when culture under non-differentiating conditions (Figure 5B and 5E). Moreover, when differentiation was induced in these deletion lines, only *RANBP9^-/-^* clones showed higher CD49d^+^/Band3^+^ populations, indicating accelerated differentiation and mirroring *UBE2H* and *MAEA* deletions (Figure 4F and 4G). We believe, that these new data support our conclusion that RANBP9- and RANBP10-CTLH complexes have functional roles at different stages of maturation, and that RANBP9-CTLH has a potential role in maintaining HUDEP2 cells in a dormant/quiescent progenitor cells and/or controlling transition to terminal differentiation. The new data are integrated and described in the revised manuscript.

2. Another conclusion that is overstated, is that RanBP9 and RanBP10 assemble in "distinct CTLH E3 supramolecular complexes" which would be of importance to support the conclusion of "structurally distinct complexes" regulating erythroid maturation. However, they did not provide evidence that RanBP9 and RanBP10 form different complexes in vivo. The authors show that RanBP9 and RanBP10 can assemble with ArmC8 and Twa1 to form CTLH complex core module in vitro, but the data from the sucrose gradients is in support of a model where RanBP9 and RanBP10 co-exist in the complex. Therefore, as stated by the authors in the discussion (line 492) "…we can only speculate about the mechanism of assembly of complex subunits", which makes it difficult to support their claim that "differential ubiquitylation modules regulate erythroid maturation".

Similar concern regarding CTLH complex modulation was raised by reviewer 1 and 3. Please see responses to reviewer 1, point 1.

3. Another main conclusion stated by the authors is that re-expression of MAEA bearing a mutation in a non-priming residue (Y394A, that they previously showed prevented ubiquitination in vitro) is not able to restore UBE2H expression, implying that the CTLH complex activity is required to maintain UBE2H expression. However, this claim is supported by one single transfection experiment, not quantified, which shows a slight reduction in UBE2H expression compared to wild-type MAEA. This is substandard. This experiment needs to be repeated and quantified. In addition, as it is known from both yeast and mammalian cell studies that MAEA and RMND5A expression are interdependent and that RMND5A expression is decreased upon MAEA knockout, it would be necessary to ensure that RMND5A expression is restored when re-expressing MAEA under the conditions used.

To strengthen our conclusion that MAEA activity is required for UBE2H levels, we generated clonal MAEA^-/-^ cell lines stably expressing either wildtype or MAEA-Y394A mutant (note, clones were selected for comparable expression levels of wildtype or MAEA-Y394A). Immunoblot analysis in revised Figure 6I confirms that MAEA-Y394A is not efficient in stabilizing UBE2H, whereas wild type completely rescued UBE2H levels. Immunoblot signals of UBE2H levels, normalized to actin, have been quantified (Figure 6J). Importantly, RMND5A levels are stabilized by wildtype MAEA expression, as well as MAEA-Y394A.

4. Adding to the above comment. This report convincingly implicates MAEA in erythropoiesis signaling, however, the connection with the CTLH complex reported by the authors is puzzling. The discussion introduces two other publications that connect MAEA and hematopoietic signaling, but the manuscript overall fails to make any other connections with the CTLH complex. What is particularly surprising is that deletion of WDR26 – a complex member that was previously shown to regulate supramolecular formation in the same cell line used in this study – is reported to have no effect on UBE2H levels compared to parental cells. The same goes for GID4, the receptor subunit of the complex, which appears to be dispensable for UBE2H regulation. Could MAEA be functioning independently of the CTLH complex to induce this phenotype? Do RMND5A knockout cells exhibit the same phenotype, since it is predicted that only the MAEA-RMND5A RING heterodimer is functional?

Structural evidences from GID/CTLH complexes show that different forms of GID/CTLH complexes can exist and active complexes without Gid4 and WDR26 also exist (Sherpa et al., 2021; Qiao et al., 2020; Mohamed et al., 2021). This might suggest why deletion of substrate receptor Gid4 or WDR26 alone does not affect UBE2H levels. However, MAEA which is the catalytic RING subunit of CTLH is indispensable for all forms of active CTLH E3 ligases. MAEA and RMND5A are structurally and functionally interconnected. As shown recently protein abundance of MAEA and RMND5A is interdependent (Maitland et al., 2019) and in agreement, we also observe in our *MAEA* deletion cells reduced amounts of RMND5A (Figure 6I)

Other Comments1. Line 197: Authors suggest that RanBP9 and RanBP10 protein levels change from day 0 to day 6 of differentiated HUDEP2 cells based off the sucrose gradient, but since there is no loading control in this type of experiment, they cannot use sucrose gradients to make this conclusion. Curiously, the authors do not see a difference in any other complex member protein levels by sucrose gradient, even though the authors conclude that there are protein level differences based on Figure 1F and G. A Western blot showing levels of complex members on day 0 and day 6, with appropriate loading controls, would allow for these conclusions while also validating the proteomic data shown in Figure 1. Also, why choose day 0 and day 6 for Figure 2 instead of day 0 and day 12? Day 12 showed the largest effects according to data in Figure 1.

Since we are looking at different maturation stages and the associated massive global protein changes, loading controls would also likely change, and therefore not be suitable. Therefore, for the sedimentation experiments equal total protein amount was loaded onto the sucrose gradient. Immunoblots (of input and fractions samples) and scans were done in parallel, hence, a relative comparison is justified.

For technical reason we did not use cell lysates from differentiation day 12. Cells at day 12 are very small and to extract amounts of protein lysates equal to cells at earlier differentiation stages (day 0 to day 8) would have resulted in an unmanageable upscaling of the differentiation protocol. Choosing time points day 0/6 for sedimentation experiments and day 0/4/8 for sequential IPs was a reasonable compromise for collecting sufficient cell lysate amounts as well as covering a time window where major CTLH remodelling occurs.

2. Line 201: ARMC8 was immunoprecipitated using a nanobody, but in the methods section (line 651), authors indicate that ARMC8 antibody was used for immunoprecipitation. This ambiguity needs to be addressed. Authors should also indicate whether the nanobody is specific for ARMC8 α or β; since this nanobody has not been used in previous publication, ARMC8 KO cell lines should be the appropriate negative control instead of "mock".

We apologize for the confusion. We generated an ARMC8-specific nanobody that was used for IPs. For the verification that the nanobody is specific for ARMC8, we resolved a cryo-EM structure of the ARMC8 nanobody bound CTLH complex (Figure 2—figure supplement 2) and the structure also indicates that the nanobody recognizes and binds a region specific to the α-variant of ARMC8. Therefore, we did not use ARMC8 KO cell lines and rather used IP in the absence of the nanobody (mock) as a control. Text and methods have been edited accordingly. Also, depletion of the core scaffolding subunit ARMC8 will disrupt the CTLH complex assembly, and thereby would be unsuitable in comparison with specific IP experiments.

3. Line 209: a couple of complex members besides RanBP9 and RanBP10 should be immunoblotted on the sucrose gradients, especially since the purpose of the figure was to demonstrate "whether RanBP9 and RanBP10 could can independently form CTLH complexes".

We expanded the immunoblot analysis and monitored WDR26 and MAEA. Both CTLH subunits co-sediment with RANBP9 or RANBP10 in respective cell lines corresponding to supramolecular assembly fractions. The new data are added in Figure 2G.

4. Line 214: authors mention that RanBP9 and RanBP10 assemble in "distinct CTLH E3 supramolecular complexes" based on Figure 2, but as seen in Figure 2A, both RanBP9 and RanBP10 exhibit a similar sedimentation pattern in parental HUDEP2 cells, suggesting that both proteins may be present in the same complex rather than RanBP9- and RanBP10-dependent complexes. Since this is a fundamental conclusion from this paper, authors need to further analyze how RanBP9 and RanBP10 interact with the complex when both are present. The authors mention (Line 495) a previous study that showed that other CTLH complex members' expression is compromised upon RanBP9 depletion, and they mention that RanBP9 is "slightly increased" in RanBP10 KO cells, although no quantifications are shown. Did the authors observe a similar decrease in CTLH complex subunits upon RanBP9 knockout as previously reported, or is there a compensatory effect in HUDEP2 cells? This would help better understand the nature of the complexes.

We thank the reviewer for pointing out the possibility of mixed RANBP9-RANBP10-CTLH complex assemblies. Indeed, the new data from the sequential IP (Figure 2D and 2E) mentioned also in response to reviewer 1, point 1, clearly show co-precipitation of RANBP10 with RANBP9 IP, indicating also presence of “mixed” RANBP9/RANBP10-CTLH complexes.

Increased RANBP10 amounts in RANBP9^-/-^ (and RANBP9 amounts in RANBP10^-/-^), have consistently been observed (Figure 2F and Figure 5A). Immunoblots do not indicate an obvious decrease of CTLH subunits like WDR26 or MAEA compared to the parental (Figure 2G). Hence, we can only speculate that the cellular stoichiometry of CTLH subunits might allow a “compensatory” assembly of RANBP9- or RANBP10-CTLH assemblies, as mentioned in the discussion.

5. Figure 3: it has previously been shown that GID4 is larger than TWA1 (Lampert et al., 2019; Qiao et al., 2020; Mohamed et al., 2021), yet Figure 3A indicates the opposite.6. Line 223: Sherpa et al., 2021 showed that when WDR26 is deleted, much of the CTLH complex population migrates to a lower molecular weight when performing sucrose gradients in vivo using K562 cells, yet the authors omitted WDR26 when reconstituting the complex in vitro. Given how supramolecular assembly, which corresponds to data shown in Figure 2A, is governed by WDR26 in human cells, there needs to be a discussion about why this protein was omitted in this experiment.

As described in the methods and Sherpa et al., 2021, a 1-99aa N-terminal deletion of GID4 has been used for reconstituting the CTLH complexes and moreover TWA1 is 2xStrep-tagged. Hence, as rightly pointed out by the reviewer, the truncated GID4 migrate at a molecular weight lower than TWA1. To make it clear, we indicated the use of truncated GID4 (∆GID4) in the figure, legend, and text.

As shown in previous studies of GID/CTLH E3 ligases (Sherpa et al., 2021; Qiao et al., 2020; Mohamed et al., 2021) multiples forms of GID/CTLH assemblies with and without Gid7/WDR26 can exist. To assess the structural architecture of RANBP10 CTLH complexes and to compare it with published RANBP9-CTLH complex, we focused on the minimal catalytically active CTLH complex composed of either RANBP9 or RANBP10, ARMC8, TWA1, ∆GID4, MAEA, and RMND5A. Hence, we omitted WDR26 in the reconstituted CTLH complex.

7. Line 478: authors mention that they have shown no differences in GID4 levels during erythropoiesis in HUDEP2 cells, but Figure 1F does not include GID4. Even though it may be in the supplemental dataset, it would be logical to include GID4 in Figure 1F as well.

Heatmaps in Figure 1 show z-score protein abundance significantly changing between different maturation stages. GID4 does not significantly change, hence it is not presented in the heat map of HUDEP2 cells (Figure 1F). In proteome data of CD34^+^ cells GID4 abundance slightly peaks at basophilic stages, but the profile is different to all the other CTLH complex subunits.

8. Line 506: the authors state: "MAEA deficiency in K562 and HUDEP2 cells caused significantly lower protein amounts of RMND5a and UBE2H…”. There is no data, or citation to be found in the manuscript that shows RMND5a depletion in HUDEP2 cells.

We thank the reviewer for pointing out the missing data. New immunoblot analysis of K562 cells, comparing parental with MAEA^-/-^ cells, show reduced protein levels of UBE2H as well as RMND5A in MAEA^-/-^ cells. The levels can be rescued by ectopic expression of wild type MAEA (Figure 6I). Figures/data are indicated and mentioned in the revised manuscript.

9. A recent paper (Zhen et al., 2020) showed that CTLH complex member mRNA levels decrease as human erythroblasts become terminally differentiated, yet Figure 1G shows the opposite effect on CTLH protein level in the same system with the notable exceptions being RanBP9 and RanBP10. Could any of your results explain why this may be?

mRNA levels are not necessarily reflecting protein amounts. Indeed, it was recently shown that dynamics of mRNA expression during terminal erythroid maturation did not accurately predict protein expression (Gautier et al. 2016). Regarding some differences of mRNA and protein levels of CTLH subunits we can only speculate that different factors like mRNA stability, mRNA maturation could account for altered translational efficiency and subsequently protein amounts. Importantly, Zhen et al., 2020 did describe inverse transcriptional profiles for *RANBP9* (down regulated) and *RANBP10* (up regulated) during terminal maturation, which agree with our proteome data. We included this observation in our discussion of the revised manuscript.

10. Line 512: is there any data from the UBE2H and/or MAEA KO proteomes compared to WT that shows decreased levels of proteins involved in maintaining a dormant/quiescent progenitor stage? The authors should analyze their datasets and address the possibility that they introduced in the discussion.

The reviewers raised an interesting point whether proteins involved in maintaining dormant/question progenitor stage might be altered in *MAEA* and *UBE2H* deletion cells. To address this question, we provide a graph with protein fold change *UBE2H^-/-^* vs parental plotted against protein fold change *MAEA^-/-^* vs parental and highlighted proteins that have been implicated in quiescence/dormancy including GATA2 (Figure 4—figure supplement 2G). However, none of these are significantly changing in *MAEA* and *UBE2H* deletion cells and are therefore unlikely the cause of the spontaneous and accelerated maturation phenotype. The point is mentioned in the discussion of the revised manuscript.

11. The model proposed in Figure 6J was not addressed in the manuscript. Is there altered hematopoiesis if RanBP9 or RanBP10 are knocked out? If the CTLH complex is going to be associated with this pathway, there needs to be evidence showing that deletion of another complex member leads to a similar phenotype.

The model figure was removed based on changes in our revised manuscript. For the answer to the altered haematopoiesis upon RANBP9 KO, please look at comment to reviewer 2, major point section 1. Also, similar phenotype is visible upon MAEA KO, where MAEA is the main catalytic subunit of CTLH complex.

Reviewer #3 (Recommendations for the authors):To address the weaknesses I outlined above, I have the following comments and suggestions for experiments:1) Differentiation stage-specific formation of RANBP9-CTHL and RANBP10-CTHL complexes:– While there is good evidence that there is differential formation of these of RANBP9-CTHL and RANBP10-CTHL during maintenance and 6 days of erythrocyte progenitor differentiation (Figure 2A,B) , it would greatly strengthen if the authors could quantify the effects. E.g. would it be possible to quantify the relative cellular amounts of RANBP9 and RANBP10 integrated into CTHL E3 at each differentiation state (input versus fraction 9,10,11 from sucrose gradients) and correlate these to the relative changes in protein levels as determined by mass spec? Alternatively the endogenous ARMC8 IPs could be quantified in the same manner. In case there is a clear correlation between the relative protein level increase and the relative integration of RANBP9 and RANBP10, such data would provide evidence that changes in CTHL composition are driven by abundance of RANBP9/10 withing cells (as suggested by the authors in the discussion). Alternatively, if there is no strict correlation, this could point to cellular mechanisms that regulate their assembly into the CTHL complex.

To address this suggestion, we set up a different approach using sequential IPs. Please see response to reviewer 1, section 1.

– To provide evidence that these differentiation-specific RANBP9-CTHL and RANBP10-CTLH are functionally distinct and important for erythropoiesis, could the authors use their available RANBP9/10 knock out cells and determine effects on progenitor maintenance and differentiation?

Please see response to reviewer 2, section 1.

2) Coupling of UBE2H stability to CTLH E3 ligase activity:– The authors convincingly show that UBE2H stability depends on the catalytic activity of the CTLH E3 ligase and in its absence is targeted for proteasomal degradation. This is an conceptually interesting finding and it would greatly strengthen the manuscript if the authors could provide evidence for when and how such regulation is important during erythrocyte progenitor maintenance or differentiation. E.g. would overexpression of UBE2H dysregulate erythrocyte progenitor maintenance, in which overall CTHL E3 ligase function is presumably lower?

We thank the reviewer for the suggestion. We are also interested in dissecting the detailed regulation of UBE2H abundance and its cellular importance, however this will require further extensive investigations beyond the scope of this manuscript and this will be our follow-up goal in future.

– Can the authors speculate on how catalytic activity of MAEA prevents proteasomal degradation of UBE2H? Could it be through ubiquitin modification itself? The activity of other E2s have been shown to be regulated by ubiquitylation (e.g. UBE2T (PMID: 16916645) or UBE2E (PMID: 25960396)), so it might be worthwhile to test whether UBE2H undergoes MAEA-dependent, regulatory ubiquitylation, which, if lost, leads to proteasomal degradation.

Regarding a potential mechanism regulating UBE2H levels in the absence of functional MAEA, we investigated whether “orphan” UBE2H might undergo autoubiquitylation. Proteomics studies identified several UBE2H ubiquitylation sites (RMIDs:29967540, 2106985, 21890473), which we highlighted in an Alphafold 2-derived model structure of UBE2H-Ub (Figure 6—figure supplement 1D). The surface-exposed sites that are in close proximity to the catalytic C87 could be potentially autoubiquitylated. We reconstituted autoubiquitylation in an in vitro assay and detected ubiquitylation of UBE2H which was dependent on the catalytic C87 residue (Figure 6K). Moreover, using lysine-less ubiquitin with all Lys-to-Arg mutation, indicated predominant UBE2H monoubiquitylation. Taken together, it is likely that in the absence of ubiquitin-transfer proficient MAEA (aka functional CTLH complex), UBE2H undergoes autoubiquitylation and subsequent proteasomal degradation. Alternatively, other UBE2H-interacting E3 ligases, such as TRIM28 (Doyle at al. 2010, PMID: 20864041), could potentially target UBE2H for proteasomal degradation.

3) CTLH E3 ligase loss and accelerated erythrocyte differentiation: while the data presented (in particular Figure 5 H) supports an accelerated differentiation phenotype, I think the authors should clarify whether they think that MAEA- and UBE2H-deficient HUDEP2 cells yield functional progenitors. At least from the proteomic data analysis in Figure 4H and J one would expect that the resulting cell populations are quite different. Could the authors perform assays to compare viability and functionality of control and CTLH E3 ligase-deficient orthochromatic erythroblasts (day 12 of differentiation)? Results from such analysis would also help understanding how erythrocyte lineage-specific Maea knock out results in reduced circulating RBC counts (PMID: 30674470) (i.e. relative contributions of early versus late defects in erythropoiesis).

Our initial in vitro erythropoiesis studies with HUDEP2 cells show spontaneous and accelerated erythroid maturation in MAEA and UBE2H deficient cells. Importantly, mouse studies, that either conditionally deleted *MAEA* in central macrophages of erythroblastic islands or in erythroid progenitors (Wei et al., 2019; PMID: 30674470), have revealed abnormal erythroblast maturation in the bone marrow showing altered profiles with distinct accumulation of maturation stages. This phenotype is, at least in part, recapitulated in our study of *MAEA-* and *UBE2H-*deficient HUDEP2 cells by an apparent accumulation of early maturation stages.

We followed up on these data and addressed reviewer’s comment regarding the functionality of *MAEA-* and *UBE2H*- deficient orthochromatic erythroblasts that might account for reduced circulating RBC counts in erythrocyte lineage-specific *MAEA* knockout (Wei et al., 2019; PMID: 30674470), or anaemia in *MAEA* null mice embryos, which also accumulate nucleated erythrocytes in peripheral blood (Soni S. et al. 2006, PMID: 16707498). The maturation of orthochromatic erythroblasts into reticulocytes is characterized by the elimination of the condensed nucleus – enucleation. Hence, we focused our functional analysis of *MAEA-* and *UBE2H*- deficient orthochromatic erythroblasts by monitoring enucleation efficiency as a critical step in generating functional reticulocytes. HUDEP2 cells show intrinsically weak enucleation efficiency, we therefore switched to the CD34^+^ erythroid maturation system. We observed inefficient enucleation of maturating CD34^+^ cell that were depleted of either *MAEA* or *UBE2H,* suggesting a role of UBE2H-CTLH at ortho/reticulocytes stage (Figure 4I-4L). To date, no UBE2H knock out mouse models and erythropoiesis studies are available, however, our data agree with data from *MAEA* null mice that accumulate nucleated erythrocytes in peripheral blood (Soni S. et al. 2006, PMID: 16707498). Cumulatively, our in vitro studies support the notion that the UBE2H-CTLH modules are required in erythroid maturation.

Additional comments:Line 507: the authors state that MAEA deficiency results in lower RMND5a amounts. I hope I am not missing something, by I could not easily see the data for this in the figures. Could the authors please clarify and cite the appropriate figure showing this?

We thank the reviewer for pointing the missing data. New immunoblot analysis of K562 cells, comparing parental with MAEA^-/-^ cells, show reduced protein levels of RMND5A in MAEA^-/-^ cells (as previously reported by Maitland et al., 2019). The levels can be rescued by ectopic expression of wild type MAEA (Figure 6I). Figure/data are indicated and mentioned in the revised manuscript.